



# Accelerating methane growth rate from 2010 to 2017: leading contributions from the tropics and East Asia

Yi Yin[a,b,1], Frederic Chevallier[b], Philippe Ciais[b], Philippe Bousquet[b], Marielle Saunois[b], Bo Zheng[b], John Worden[c], A. Anthony Bloom[c], Robert Parker[d], Daniel Jacob[e], Edward J. Dlugokencky[f], and Christian Frankenberg[a,c]

[a]Division of Geological and Planetary Sciences, California Institute of Technology, Pasadena, CA, USA
[b]Laboratoire des Sciences du Climat et de l'Environnement, CEA-CNRS-UVSQ, Gif-sur-Yvette, France
[c]Jet Propulsion Laboratory, California Institute of Technology, Pasadena, CA, USA
[d]National Centre for Earth Observation, University of Leicester, Leicester, UK
[e]School of Engineering and Applied Sciences, Harvard University, Cambridge, MA, USA
[f]NOAA Earth System Research Laboratory, Boulder, Colorado, USA

**Correspondence:** Yi Yin (yiyin@caltech.edu)

**Abstract.** After stagnating in the early 2000s, the atmospheric methane growth rate has been positive since 2007 with a significant acceleration starting in 2014. While causes for previous growth rate variations are still not well determined, this recent increase can be studied with dense surface and satellite observations. Here, we use an ensemble of six multi-tracer atmospheric inversions that have the capacity to assimilate the major tracers in the methane oxidation chain – namely methane, formaldehyde, and carbon monoxide – to simultaneously optimize both the methane sources and sinks at each model grid. We show that the recent surge of the atmospheric growth rate between 2010-2013 and 2014-2017 is most likely explained by an increase of global $CH_4$ emissions by $17.5\pm1.5$ Tg yr$^{-1}$ (mean$\pm1\sigma$), while variations in $CH_4$ sinks remained small. The inferred emission increase is consistently supported by both surface and satellite observations, with leading contributions from the tropics wetlands ($\sim$35%) and anthropogenic emissions in China ($\sim$20%). Such a high consecutive atmospheric growth rate has not been observed since the 1980s and corresponds to unprecedented global total $CH_4$ emissions.

## 1 Introduction

Methane ($CH_4$) is an important greenhouse gas highly relevant to climate mitigation, given its stronger warming potential and shorter lifetime than carbon dioxide ($CO_2$) (IPCC, 2013). Atmospheric levels of methane, usually measured as dry air mole fraction [$CH_4$], have nearly tripled since the Industrial Revolution according to ice core records (Etheridge et al., 1998; Rubino et al., 2019). This increase is mostly due to increases in anthropogenic emissions from agriculture (ruminant livestock and rice farming), fossil fuel use, and waste processing (Kirschke et al., 2013; Saunois et al., 2016; Schaefer, 2019). A significant portion of methane is also emitted from natural sources, including wetlands, inland freshwaters, geological sources, and biomass





burning (although many of the wildfires may have anthropogenic origins) (Saunois et al., 2016). Methane has a lifetime of around 10 years in the atmosphere (Naik et al., 2013), with a dominant sink from oxidation by hydroxyl radicals (OH) in the troposphere (∼90% of the total sink) (Saunois et al., 2019)). Besides, its reactions with atomic chlorine (Cl), soil deposition, and stratospheric loss through reaction with a range of reactants (including O(‘D), Cl and OH) account for a minor portion of the total methane sink (Saunois et al., 2019).

Since the beginning of the direct measurement period in the early 1980s, [$CH_4$] growth rate had been gradually declining until it reached a stagnation between the late 1990s and 2006, often referred to as the "stabilization" period (Dlugokencky et al., 1998, 2003). However, [$CH_4$] has been increasing again since 2007 (Dlugokencky et al., 2009; Nisbet et al., 2014). A sharp increase of the growth rate was observed in 2014 from surface background stations (12.6±0.5 ppb $yr^{-1}$, mean ± 1 $\sigma$) (Nisbet et al., 2016; Fletcher and Schaefer, 2019; Nisbet et al., 2019), more than twice the average growth rate of 5.7±1.1 ppb $yr^{-1}$ during the post stagnation period between 2007 and 2013. Since then, the $CH_4$ growth rate has remained high (8.6±1.6 ppb $yr^{-1}$ for 2014-2017). Understanding methane source and sink changes underlying these [$CH_4$] variations can help us identify how methane sources respond to human activity, climate, or environmental changes, which are critical to climate mitigation efforts.

The attribution of the plateau and regrowth in [$CH_4$] during the 2000s reached conflicting conclusions about the role of fossil fuel emissions (Hausmann et al., 2016; Simpson et al., 2012; Worden et al., 2017), agriculture or wetland emissions (Nisbet et al., 2016; Saunois et al., 2017; Schaefer et al., 2016), OH concentration (Rigby et al., 2017; Turner et al., 2017), and biospheric sinks (Thompson et al., 2018). The range of competing explanations exemplifies the complexity and uncertainty of interpolating limited observations of [$CH_4$] and the $^{13}C/^{12}C$ isotopic ratio (expressed as $\delta^{13}CH_4$) to changes in different sectors of methane sources as well as its sinks (Turner et al., 2019; Schaefer, 2019). The situation now is more encouraging than the previous decade as we have continuous global satellite retrievals of the total column $CH_4$ dry air mole fraction (denoted as $X_{CH_4}$) from the Greenhouse Gases Observing Satellite (GOSAT) with better precision and accuracy than previous instruments (Kuze et al., 2009; Parker et al., 2015; Jacob et al., 2016; Buchwitz et al., 2017; Houweling et al., 2017). The combined information from satellite and surface observations – the latter with the largest networks of surface stations so far in measurement history – provides us a unique opportunity to understand the recent changes in [$CH_4$] with better spatial coverage.

Atmospheric [$CH_4$] measurements can be linked quantitatively to regional sources and sinks by inverse modeling, where changes in the atmospheric transport are guided by meteorological reanalysis and fluxes are adjusted to match the temporal and spatial variations of the observations given their uncertainties in a Bayesian formalism (Chevallier et al., 2005). A number of inverse studies have explored the surface and GOSAT observations to improve methane emission estimates (Monteil et al., 2013; Cressot et al., 2014; Alexe et al., 2015; Miller et al., 2019; Ganesan et al., 2017; Maasakkers et al., 2019), but the recent acceleration of [CH4] growth since 2014 has not been widely investigated (Nisbet et al., 2019). GOSAT satellite $X_{CH_4}$ retrievals agree with the surface [$CH_4$] observations on the acceleration of the increase in atmospheric methane burden over the period from mid-2009 to the end of 2017 (Fig. 1). However, the satellite column data show a smoother temporal variation in





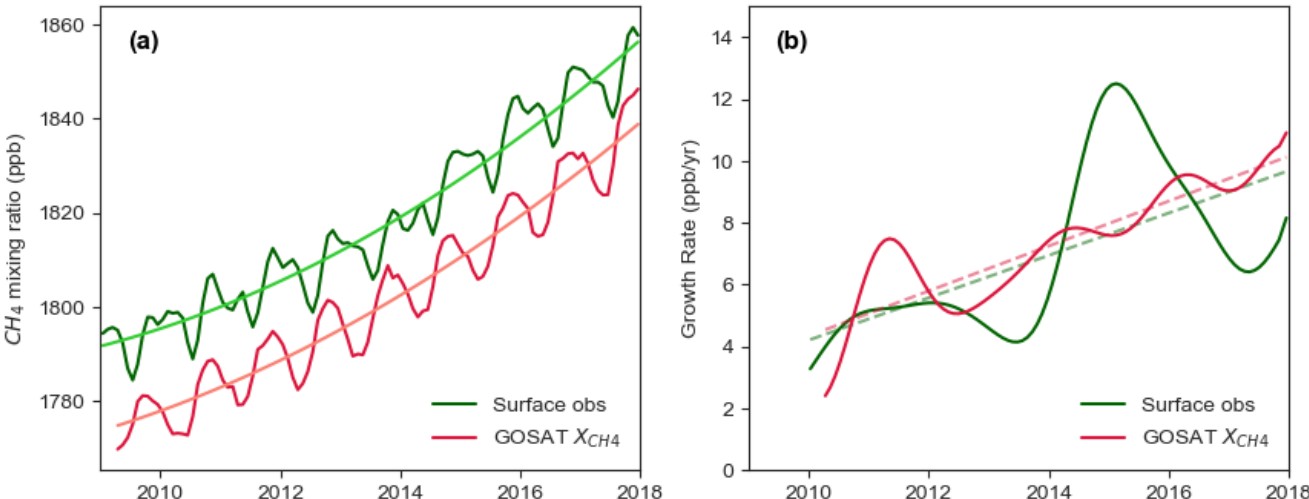

**Figure 1.** Atmospheric methane mixing ratio changes. (a) Monthly time series of the global mean methane mixing ratio from mid-2009 to the end of 2017. The green curve represents [CH$_4$] in the marine boundary layer observed by the NOAA surface network (www.esrl.noaa.gov/gmd/ccgg/mbl/). The red curve represents the total column mixing ratio, X$_{CH4}$, seen by GOSAT satellite and averaged from all soundings over the land. The smooth curve fit shows a quadratic fit of the trend that accelerates in the latter part of the study period. (b) Smooth methane growth rate derived from the time series as shown in (a) following methods of (Thoning et al., 1989).

the global average growth rate. GOSAT X$_{CH_4}$ growth rates over different regions show diverse temporal patterns with a higher variability than the global average (Fig. S1), suggesting that satellite data sampling directly over the source regions could provide valuable information to track regional changes in CH$_4$ fluxes. Furthermore, species in the oxidation chain of methane, namely methane-formaldehyde-carbon monoxide (CH$_4$-HCHO-CO) with their reactions to OH as the common sink path, could

provide additional constraints on the OH sink of methane. Recent study has shown that HCHO levels can inform about remote tropospheric OH concentrations (Wolfe et al., 2019), and the feedback of CO variations on OH is directly linked to the sink of CH$_4$ (Gaubert et al., 2017; Nguyen et al., 2020). Hence, satellite retrievals of X$_{HCHO}$ from the Ozone Monitoring Instrument (OMI, (González Abad et al., 2016)) and X$_{CO}$ from the Measurements of Pollution in the Troposphere (MOPITT, (Deeter et al., 2017)) covering the study period could, in theory, provide additional constraints on regional variations of methane sinks.

Hence, we developed a multi-tracer variational inverse system, PYVAR-LMDZ, with the capacity to assimilate observations of the CH$_4$-HCHO-CO oxidation chain to better constrain the sources and sinks of these species at individual model grid cell (Chevallier et al., 2005; Pison et al., 2009; Fortems-Cheiney et al., 2012; Yin et al., 2015; Zheng et al., 2019). Given observed changes in temporal and spatial variations of all the three species, we optimize simultaneously (i) methane emissions, (ii) CO emissions, (iii) HCHO sources (surface emissions + chemical productions from VOC oxidation), and (iv) OH

concentrations. These terms are optimized at a weekly temporal resolution and a 1.9° by 3.75° spatial resolution. Besides, we optimize the initial concentrations of all the four species at individual horizontal model grid. Here, we performed an ensemble



of six inversions using different combinations of observational constraints (surface vs. satellite, single vs. multiple species) and alternative prior estimates of 3-D OH distributions. With the ensemble results, We aim to (1) identify key regions that contribute to the [CH$_4$] growth rate acceleration from 2010 to 2017, and (2) evaluate the consistency of results inferred from surface and

satellite observations. Inversion methods and observational datasets are documented in Section 2. We report estimates of global methane budget change from 2010 to 2017 in Section 3 and discuss regional attributions and sources of uncertainties in Section 4. Section 5 summarizes this work and provides some perspectives for future studies.

## 2 Data and Methods

### 2.1 Atmospheric Observations

We assimilate surface and satellite [CH$_4$] observations in parallel to test the consistency of information brought by these two types of measurements. We also include versions assimilating HCHO and CO along with CH$_4$ to test the impacts of adding chemically related species. In total, there are three groups of observational constraints:

-**S1**: Surface [CH$_4$] and [CO] measurements;

-**S2**: GOSAT X$_{CH_4}$;

-**S3**: GOSAT X$_{CH_4}$, OMI X$_{CH_2O}$, and MOPITT X$_{CO}$. The assimilation is done from April 2009 to February 2018, and we analyze the results of the eight full years of 2010-2017 with the starting and ending period being spin-up and spin-down phases to avoid edge effect.

### 2.1.1 Surface Observations

We include surface [CH$_4$] from a total of 103 stations (Fig. S2; Table S3), with leading contributions from the following net-
works: U.S. National Oceanic and Atmospheric Administration (NOAA, 58 stations), Australia's Commonwealth Scientific and Industrial Research Organisation (CSIRO, 9 stations), Environment and Climate Change Canada (ECCC, 8 stations), and AGAGE (5 stations, (Prinn et al., 2018)). Measurements from different networks are calibrated to the WMO scale. Daily afternoon averages between 12 and 6 pm local time are used for the assimilation of the continuous in-situ measurements to minimize uncertainties associated with boundary layer height modeling. CO observations from those stations are also assimilated in **S1**.

### 2.1.2 Satellite Observations


The TANSO-FTS instrument onboard the Greenhouse Gases Observing Satellite (GOSAT) was launched by The Japan Aerospace Exploration Agency (JAXA) into a polar sun-synchronous orbit in early 2009. It observes column-averaged dry-air carbon





dioxide and methane mixing ratios by solar backscatter in the shortwave infrared (SWIR) with near-unit sensitivity across the air column down to surface (Butz et al., 2011; Kuze et al., 2016). Observations are made at a local time around 13:00 with a

circular pixel of around 10km in diameter. The distances between pixels both along and cross track are ∼250 km in the default observation mode, and the revisit time for the same observation location is 3 days. Denser observations over particular areas of interest are made in target mode. Here, we use GOSAT $X_{CH_4}$ proxy retrievals (OCPR) version 7.2 from the University of Leicester, which has been well documented and evaluated against various observations, with a single-sounding precision of ∼0.7% (Parker et al., 2015). This product is consistent with other GOSAT methane retrievals (Buchwitz et al., 2017). We only

assimilate GOSAT retrievals over land to minimize potential retrieval biases between nadir and glint viewing modes. The same GOSAT data are assimilated in both **S2** and **S3**.

For the multi-tracer inversion, **S3**, we also include OMI $X_{HCHO}$ retrievals version 3 from Smithsonian Astrophysical Observatory (SAO) (González Abad et al., 2016) and MOPITT $X_{CO}$ retrievals version 7 from NCAR (Deeter et al., 2017). All satellite retrievals are processed following the recommend quality flags and the application of corresponding prior profiles and

retrieval averaging kernels when provided. We exclude data poleward of 60°. Individual retrievals that are located in the same model grid within 3-hour intervals are averaged for further assimilation. The observation uncertainty contains the retrieval errors as reported by the data product plus model errors whose standard deviations are empirically set as 1% for $CH_4$, 30% for CO, and 30% for HCHO based on previous experiments (Fortems-Cheiney et al., 2012; Cressot et al., 2014; Yin et al., 2015).

### 2.1.3   Ground-based total column measurements

Ground-based $X_{CH_4}$ retrievals from the Total Carbon Column Observing Network (TCCON) from 27 stations are used for an independent evaluation of the posterior model states. TCCON is a network of Fourier transform spectrometers (FTSs) from near-infrared (NIR) solar absorption spectra, designed to retrieve precise total column abundances of $CO_2$, $CH_4$, $N_2O$ and CO to validate satellite observations (Wunch et al., 2011).

## 2.2   Inverse Modeling

### 2.2.1   Variational Inverse System

We use a Bayesian variational inversion system, PYVAR-LMDz, which uses LMDz-INCA as the chemistry transport model (CTM) (Hourdin et al., 2013; Hauglustaine, 2004). This inversion system has been documented and evaluated by a series of studies focusing on tracers including $CH_4$ (Pison et al., 2009; Locatelli et al., 2015; Cressot et al., 2014), HCHO (Fortems-Cheiney et al., 2012), CO (Yin et al., 2015; Zheng et al., 2019), and $CO_2$ (Chevallier et al., 2005, 2010). Here, we use a recently





developed version that has the capacity to assimilate observations of the major tracers in the $CH_4$ oxidation chain, namely $CH_4$-HCHO-CO, with OH being their common sink path, to optimize the sources and sinks for all these species simultaneously.

The CTM version we use here has a horizontal resolution of 1.875°×3.75° (latitude, longitude) and a vertical resolution of 39 eta levels. Atmospheric transport is guided by meteorological reanalyses (Dee et al., 2011) to represent changes in the dynamics. Given observational information of the three species, the system optimizes the following quantities at each grid cell

at a weekly resolution: (i) surface emissions of $CH_4$, (ii) surface emissions of CO, (iii) scaling factors for the sum of HCHO emissions and its chemical production from hydrocarbon oxidation, (iv) scaling factors of the OH concentration, and (v) the initial state of all the four species $CH_4$, HCHO, CO, and OH. The assimilation is performed continuously for the entire study period to avoid errors in temporal segmentation. The minimization of the cost function is solved iteratively until it reaches a reduction of 99% in the gradient of the cost function or a minimum of 45 iterations.

### 2.2.2 Prior estimates of surface methane fluxes and OH Fields

We use prior estimates of climatological methane emissions from various sectors except for biomass burning. This choice is made to avoid prior assumptions about the interannual variations (IAV) or trends in the surface emissions so that IAV in the posterior fluxes are primarily driven by assimilated observations. The exception made for fire emissions is due to their non-Gaussian distribution and large variations across different seasons and years where the bottom-up estimates based on

satellite-derived burned areas bring valuable prior information to guide the solution. The emission datasets from different sectors are listed in Table S2, and their spatial distributions are shown in Fig. S3. Note that soil deposition is treated as negative fluxes from the land to the atmosphere, and the emissions reported in this study are hence the net methane fluxes from the land to the atmosphere. The Gaussian uncertainty is set as 70% and 100% respectively for gridded $CH_4$ and CO emissions, whereas 200% for chemical HCHO productions and 20% for OH. Those errors are chosen empirically given the spreads across

different bottom-up estimates. The a priori spatial error correlations are defined by an e-folding length of 500 km over the land and 1000 km over the ocean. Temporal error correlations are defined by an e-folding length of 2 weeks. We do not account for error correlations across species.

We include two alternative prior estimates for the OH concentration: one based on a full chemistry simulation by the model LMDZ-INCA (Hauglustaine, 2004), noted as INCA-OH hereafter, and one from the TransCom model intercomparison

experiment for methane and related species (Patra et al., 2011), noted as TransCom-OH. The two OH fields have contrasting 3D distributions that could help to evaluate the impact of OH distributions on the resultant methane fluxes (Yin et al., 2015). In particular, the two OH fields have different Northern to Southern hemisphere ratios: ∼1.2 for INCA and ∼1 for TransCom. Similar to the prior estimates of the emissions, there are no interannual variations in the prior estimates of OH fields. Note that for the case of assimilating surface observations (S1), the spatial error correlation of OH are set to 1 within 6 latitudinal zones





(90-60S, 60-30S, 30-0S, 0-30N, 30-60N, 60-90N) and 0 across them, i.e. the zonal mean OH is optimized instead of per grid cell given limited observational constraints.

In summary, we include six inversions here with three different observational constraints and each pairing with two different prior estimates of global OH distributions (Table S1).

### 2.2.3 Information Content Analysis

While the variational inverse system has the advantage of optimizing large state vectors of fluxes for multiple species at high spatial and temporal resolutions, it is computationally too expensive to calculate the error covariances of posterior fluxes. Hence, we perform additional analytical inversions for aggregated source regions to estimate information content of available [CH$_4$] observations on regional emissions and posterior error covariances. In this configuration, the state vector $x$ becomes monthly regional emissions from 18 regions across the globe (regional mask shown in Fig. 8) plus a background term and

the impacts of changes in OH are not accounted for. The transport model and observation operator $K$, relating each element of $x$ to observable quantities $y$ can be numerically simulated. Using $x_a$ to represent the prior, $S_a$ and $S_\varepsilon$ to represent the error covariance matrices of the state vector $x$ and of the observation vector $y$, the a posteriori solution is expressed as

$$\hat{x} = x_a + G(y - Kx_a) \tag{1}$$

where

$$G = S_a K^T (K S_a K^T + S_\varepsilon)^{-1} \tag{2}$$

Here, $G$ represents the gain matrix that describes the sensitivity of the fluxes to observations, i.e. G=$\partial\hat{x}/\partial y$. The error covariance matrix $\hat{S}$ of $\hat{x}$ can be derived as

$$\hat{S} = (K^T S_\varepsilon^{-1} K + S_a^{-1})^{-1} \tag{3}$$

The ability of an observational system to constrain the true value of the state vector can be represented by the sensitivity of the

posteriori solution $\hat{x}$ to the true state $x$, commonly termed as the averaging kernel matrix $\mathbf{A}=\partial\hat{x}/\partial x$, as the product of the gain matrix $G$ and the Jacobian matrix $\mathbf{K}=\partial y/\partial x$, so that $\mathbf{A=GK}$ (Rodgers and D., 2000). This complementary analysis provides us important estimates of how much information content can the surface and satellite [CH$_4$] observations provide on regional methane emission changes.



## 3 Changes in the global CH$_4$ budget from 2010 to 2017

### 3.1 Changes in [CH$_4$] growth rate

The posterior model states generally capture well the global average [CH$_4$] growth rate both at the boundary layer and through the total column, irrespective of which data being assimilated (Fig. 2b and c). Sampled from the same ensemble of posterior model states, the surface growth rates show a sharp increase in 2014 (Fig. 2b), whereas more gradual increase is found in the column average (Fig. 2c). The agreement across different inversions demonstrates that differences in the temporal variations of the growth rates seen by surface and GOSAT observations are primarily due to 3-dimensional sampling differences rather than by some inconsistency between those two types of observations. This contrast suggests that the sharp increase in the surface [CH$_4$] growth rate in 2014 could have been amplified by sampling effect of the sparse surface network as also shown by a longer record (Pandey et al., 2019). Surface in-situ observations with high precision and accuracy provide critical anchoring points for monitoring the global background CH$_4$ concentrations in the boundary layer, while satellite retrievals are sensitive to the entire atmospheric column filling in continental gaps that are not effectively covered by surface stations. The consistency between the two observation approaches demonstrates a robust constraint on the acceleration of the atmospheric growth rate at the global scale.

### 3.2 Changes in global CH$_4$ emissions

Posterior global CH$_4$ emissions derived from all the six inversions show similar inter-annual variations (IAV) regardless of which observations are assimilated or which prior OH fields are used (Fig. 2a). As stated in the method, the prior CH$_4$ emission IAV only accounts for fire emissions, while the other emission sectors are represented by climatological means, hence the IAV of the posterior emissions are primarily driven by [CH$_4$] observations. Surface and satellite observations derive generally consistent IAV results. The choice of the prior OH fields has a notable effect on the magnitude of the optimized global emissions but not on the inferred temporal changes. Inversions using INCA-OH derives on average $20 \pm 1.5$ Tg yr$^{-1}$ higher emissions due to a larger OH sink (higher Northern Hemisphere OH concentrations). Therefore, in this study, we focus primarily on the IAV of methane fluxes that are directly relevant to changes in the [CH$_4$] growth rate while avoiding systematic differences across different inversions.

Global CH$_4$ emissions increased by $17.5 \pm 1.5$ Tg yr$^{-1}$ between 2010-2013 and 2014-2017 (the uncertainty range represents the standard deviation of the six inversions throughout this study). On average, the increase amounts to a linear trend of $4.1 \pm 1.2$ Tg yr$^{-2}$ over the eight years, corresponding to nearly a 1% increase per year. The lowest annual total emission occurred in 2012 and the highest in 2017. Current global CH$_4$ emissions are thus at a maximum level within the past million years, with high growth rates similar to the 1980s, during which the total methane loss rate was, however, not as high as today due to a lower CH$_4$ burden.

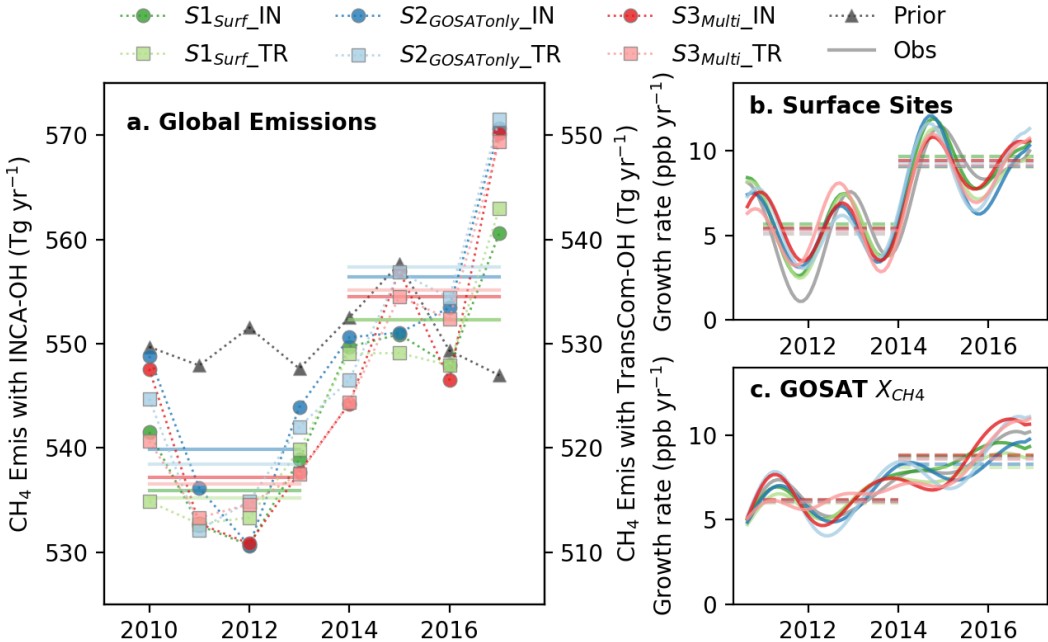

**Figure 2.** Global $CH_4$ emissions and atmospheric growth rates from 2010 to 2017. (a) Surface $CH_4$ fluxes of the prior (black triangles) and posterior estimates (color-coded). The circles represent versions using INCA-OH (denoted with IN as suffix), referring to the y-axis on the left while the squares represent versions using TransCom-OH (denoted with TR as suffix), referring to the y-axis on the right, which has a -20 Tg shift relative to the y-axis on the left. The horizontal lines mark the average emissions of the two periods, 2010-2013 and 2014-2017. (b) Deseasoanlized [$CH_4$] growth rates smoothed for variations shorter than 90 days in the posterior model states sampled at the 103 surface stations included in inversion S1. (c) $X_{CH_4}$ growth rates in the posterior model states sampled at the measurement time and location of GOSAT retrievals included in inversions S2 & S3.

## 3.3 Variations attributed to OH

Changes in the inferred OH concentrations are less than 1% at the global scale, with a small increase during 2010-2014 followed by a small decline thereafter (Fig. 3). The resulting decrease in OH since 2014, albeit small in magnitude, occurs in both the surface-driven (S1) and satellite-driven (S3) inversions, most notably in the Southern Hemisphere (Fig. S4). Inflating the prior OH uncertainty up to ±50% at each model grid only results in larger scaling factors on the OH distribution but not higher temporal variations. Such small interannual variations in the posterior OH field is consistent with a high OH recycling proba-

bility, i.e. a weak sensitivity to emission perturbations (Lelieveld et al., 2016). Some atmospheric chemistry models simulate a slightly larger year-to-year variability (1-4%) (Holmes et al., 2013; Turner et al., 2018), while recent data-constrained estimates using observed ozone columns, water vapor, methane, model-simulated NOx, and Hadley cell width suggest a relatively stable OH level over the past several decades (Nicely et al., 2018). In addition, compared to earlier box model studies that infer around 5% OH IAV from methyl chloroform (MCF) and $\delta^{13}CH_4$ observations (Turner et al., 2017; Rigby et al., 2017), a recent box



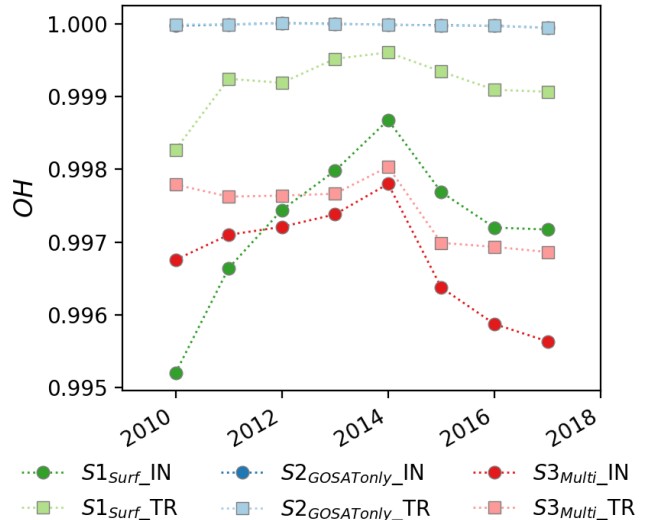

**Figure 3.** Global average posterior scaling factors on OH. Note that OH is only optimized by the system if other tracers in addition to CH4 are assimilated (S1 & S3).

model study that accounts for model biases related to tracer specific dynamics suggest a smaller IAV in OH (Naus et al., 2019). Still, we cannot rule out the possibility that our numerical optimization system preferably adapts short-term emissions to fit the observations rather than modifying OH to adjust the methane lifetime in the absence of a mechanistic chemical feedback in our chemistry-transport model (Prather, 1994; Turner et al., 2019; Nguyen et al., 2020). We will further discuss OH-related uncertainties when presenting regional results below.

# 4 Regional contributions

## 4.1 Changes in zonal CH$_4$ emissions

Similar zonal emission increases between 2010-2013 and 2014-2017 are found across the six inversions (Fig. 4a), even though they produce different latitudinal distributions of CH$_4$ fluxes (Fig. 4b). Both satellite and surface data suggest that the largest increase occurred in the southern tropics (0-30°S, 7.5±2.1 Tg yr$^{-1}$) and the northern mid-latitudes (30-60°N, 6.5±0.8 Tg yr$^{-1}$, while a moderate increase is found in the northern high latitudes (60-90°N, 1.3±0.5 Tg yr$^{-1}$). For the northern tropics (0-30°N), most versions suggest a small increase, but one version assimilating surface data suggests a small decline. Different versions agree on the overall spatial distribution of the inferred emission trends, with the most significant increase seen in East China, tropical South America, tropical Africa, and Russia (Fig. 5). Opposing trends are noted in Indochina and Southeast Asia that result in more divergent estimates across the different inversions in the 0-30°N zone.





**a.**

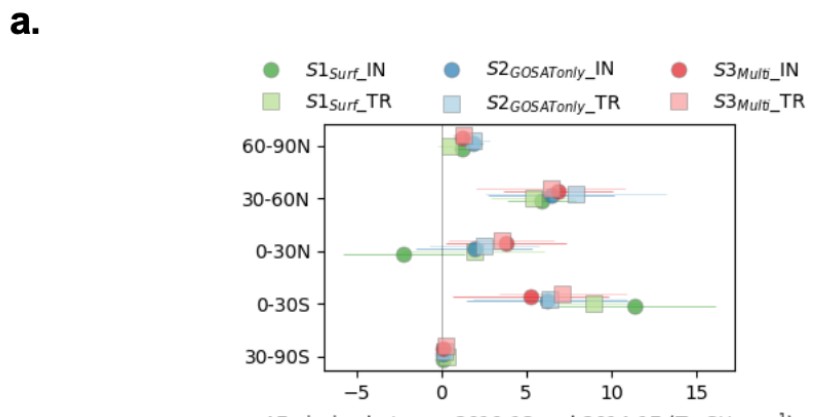

**b.**

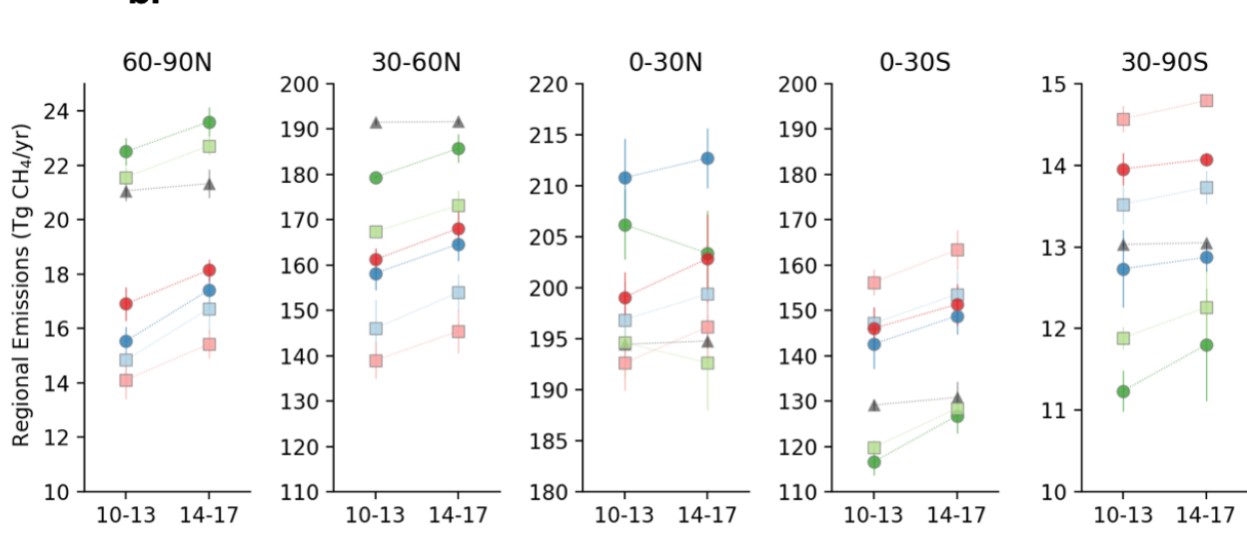

**Figure 4.** (a) Emission change between 2010-2013 and 2014-2017 in the five latitudinal zones. The error bars represent the standard deviation of changes in CH4 fluxes between the two periods. (b) Zonal fluxes estimated by different versions for the period 2010-2013 and 2014-2017. The mean values for each 4-year period are shown and errors bars represent their 1-sigma standard deviations.





Differences of zonal flux distributions are noted across versions, most notably between surface and satellite data constraints. For the same observational constraints, inversions using INCA OH fields result in higher Northern hemisphere emissions compared to the cases using TransCom OH fields due to a higher North-to-South Hemispheric OH ratio of the former. Compared to the results assimilating surface observations (S1), assimilating GOSAT $X_{CH_4}$ retrievals (S2 & S3) allocates smaller emissions in the Northern mid- and high-latitude (30-60°N and 60-90°N) but higher emissions in the tropics and

subtropics (0-30°N and 0-30 °S) (Fig. 4b). Such difference is, to a large extent, related to a latitudinal-dependent difference between model states that fit surface data and that fit GOSAT data. Specifically, the posterior model states of S1 that fit surface observations show positive biases against GOSAT $X_{CH_4}$ in the Northern mid-high latitudes but negative ones in the tropics (Fig. S5). Symmetrically, the posterior model states of S2 and S3, which fit GOSAT $X_{CH_4}$ well, show negative biases in the Northern mid-high latitudes against surface observations (Fig. S7), while the biases turn positive gradually toward the tropics

and the southern hemisphere (Figure S16). However, no latitude-dependent biases are found between GOSAT-assimilated posterior model states (S2 & S3) against TCCON total column measurements, and the magnitude of remaining biases are in line with GOSAT data validation (Parker et al., 2015). Yet S1 show similar model bias structure against TCCON as compared to GOSAT $X_{CH_4}$ (Fig. S7), suggesting discrepancies in the vertical distribution of [CH$_4$] between the model and the total column observations. Such a bias pattern between model and surface or GOSAT data has been identified by previous inverse studies

(Alexe et al., 2015; Turner et al., 2015; Miller et al., 2019; Maasakkers et al., 2019), which is likely related to biases in the model representation of the stratosphere. An empirical bias-correction on the GOSAT data so that the assimilated model states also agree with surface observations are typically applied by some studies. Here, since we focus on the IAVs of the posterior fluxes where systematic biases do not impact such results, we did not apply an empirical bias correction to the GOSAT data. Future studies to correct those biases with mechanistic understandings are needed.

**4.2    Information content of observations on regional fluxes**

To assess the extent to which the surface and satellite observations can inform us about changes of methane fluxes in distinct regions, we conducted an information content analysis for a total of 18 regions (see Section 2.2.3). The regional mask following the convention of the Global Carbon Project (Saunois et al., 2019) is shown in Fig. 8. Note that this analysis assumes all [CH$_4$] changes are resulted from surface flux changes and hence does not account for potential contributions from changes in OH

or other sink processes. The results suggest that, in most cases, GOSAT data provide more constraints on regional emissions than the surface observations (Fig. 6). This is particularly obvious in the tropics and subtropics, including Amazon, Eastern Brazil, Southern South America, Northern Africa, Tropical Africa, Southern Africa, Mideast, India, and Southeast Asia. This is because fewer surface sites exist in those regions but satellite data have a better coverage. Consequently, the posterior errors in the optimized emissions constrained by satellite data are less correlated across different regions compared to the case with

surface data constraints only (Fig. S8). The error covariances suggest that the surface observations alone, mostly located in the background boundary layer, is insufficient to separate tropical emissions from the three continents – South America, Africa, and

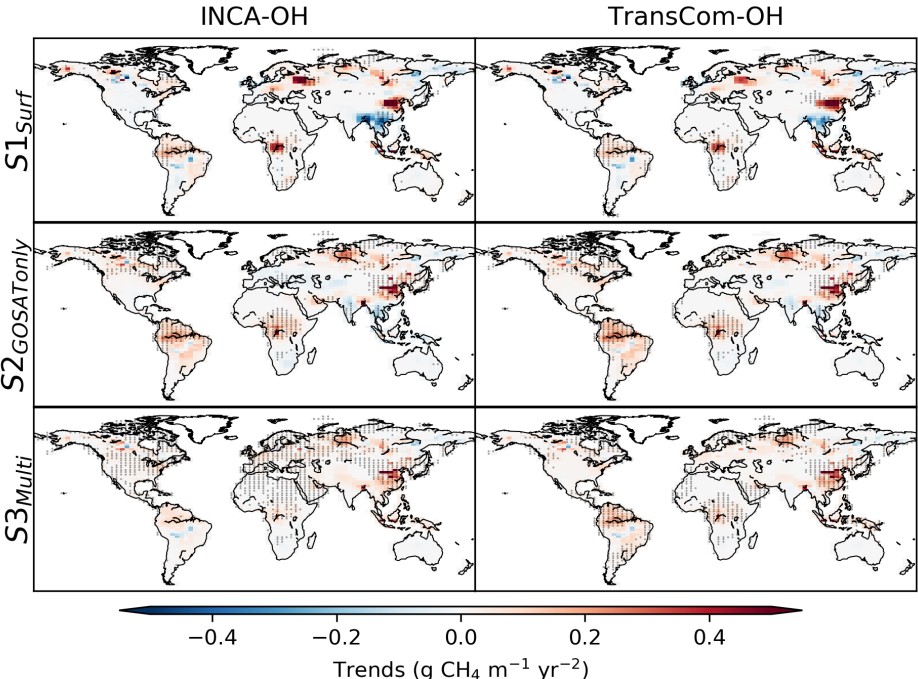

**Figure 5.** Spatial distribution of trends in the posterior CH$_4$ emissions from 2010 to 2017. The left column shows results using INCA-OH and the right column uses TransCom-OH. Each row represents one type of observational constraints.

Asia. In contrast, the cross-error terms in the GOSAT inversion are much smaller, suggesting that to a large extent emissions from different regions can be individually constrained by these X$_{CH_4}$ observations.

### 4.3 Regional emission changes

Breaking down changes in the posterior CH$_4$ emissions between 2010-2013 and 2014-2017 into the 18 regions, the most substantial increases occurred in Amazon, China, and Tropical Africa, by 4.2±1.2, 3.7±1.0, and 2.1±0.8 Tg yr$^{-1}$ respectively (Fig. 7). Changes in the three regions amounts to nearly 60% of the global emission increase. While all the six inversions agree on such a regional pattern, the multi-tracer versions (S3), that optimize OH concentrations simultaneously with the surface methane fluxes, infer smaller CH$_4$ emission increases compared to the version assimilating GOSAT alone (S2). This

difference could stem from adapting the regional mean OH level that converts the same concentration change to different emission changes. In addition, differences between S2 and S3 could result from the variational inversion reaching different approximations of the cost function minimum. For the leading contributing regions, we note a general increase in the gradient of X$_{CH_4}$ between the source regions where we find major increases and the remote ocean along the same latitudes across the study period, even though there are considerable uncertainties associated with sampling, data gaps, and atmospheric transport





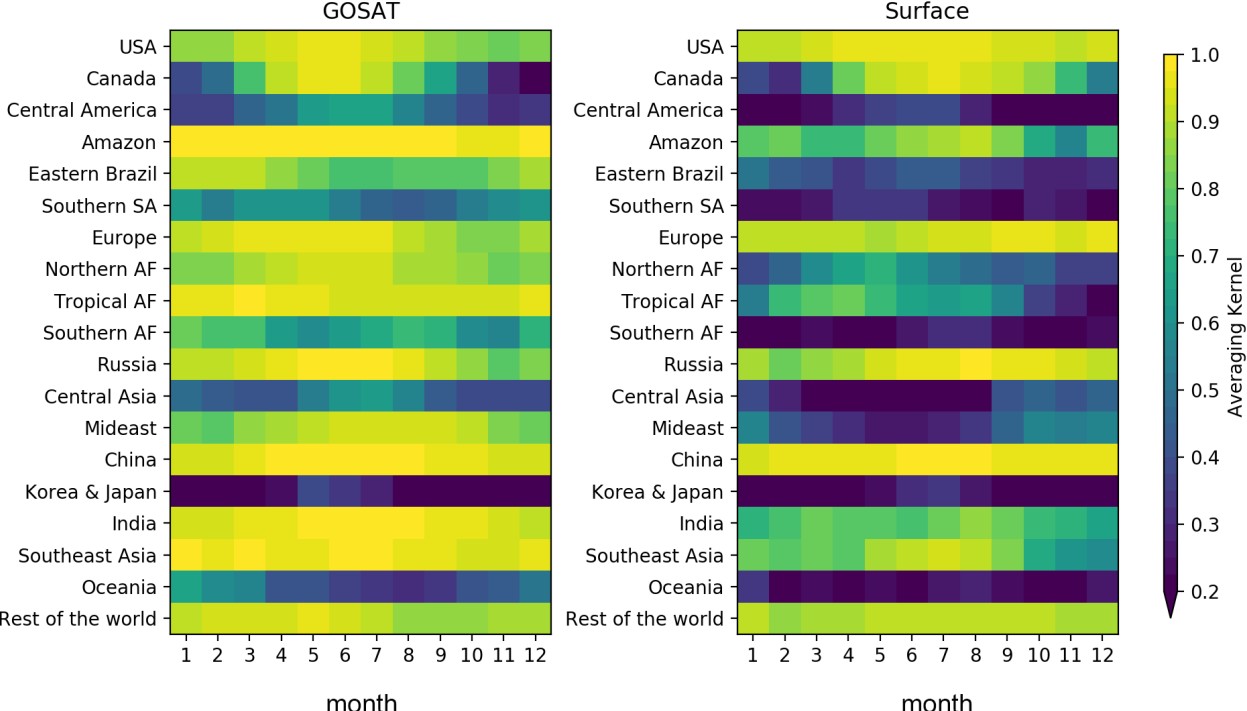

**Figure 6.** Averaging kernels (AK) of regional emissions to observations over that region during each month. Results for the year 2010 is shown here as an example.

(Fig. S9). This temporal pattern supports the interpretation of changes primarily in the surface sources rather than in the atmospheric sink as the influence of the latter on $X_{CH_4}$ would be mixed zonally.

To gain further understanding of observed changes in regional $CH_4$ emissions, we attribute our inversion emission anomaly estimates into the following categories, based on our prior bottom-up emission inventory: fossil fuel (oil, gas, coal mining, industry, residential, transport, and geological), waste (landfills and wastewater), agriculture (enteric fermentation,

manure management, and rice cultivation), wetlands (including inland water), and fire (including biofuel). We acknowledge the fact that this prior information has significant uncertainties as evidenced by the large spread across different bottom-up inventories (Saunois et al., 2016). The proportion of the different sectors remains unchanged in each grid cell throughout all years, except for fire, because we use a climatological estimates for prior emissions. Our emission attribution thus reflects a likelihood of contributing processes at a given location and season, which is larger, and most useful, in regions where emissions

are predominately contributed by a specific sector (Fig. S10 & S11).

For the Amazon, wetlands are the major contributor to $CH_4$ emissions according to the bottom-up emission inventories, and hence our identified source for the increase, showing an average trend of $0.8\pm0.1$ Tg yr$^{-2}$ over the eight study years with

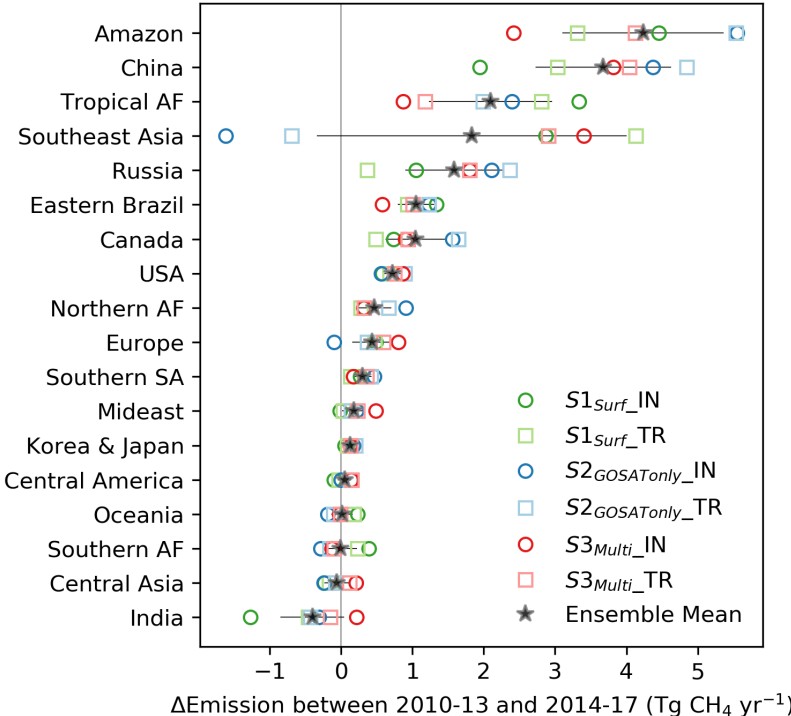

**Figure 7.** Regional emission changes between 2010-13 and 2014-2017 ranked from the highest to the smallest changes. The color-coded markers represent individual inversions, the grey stars represent the ensemble mean, and the horizontal error bar denotes the standard deviation of all versions. The regional mask is shown in Fig. 8.

shorter-term interannual variations (Fig. 8). Fire emissions from this region were high during the 2010 drought but did not rise significantly in the recent 2015 El Nino, in agreement with previous estimates based on CO and $CO_2$ (Gatti et al., 2014; Liu et al., 2017). No significant trend in the anthropgenic emissions are noted up to 2014 according to the most recent updates from the Community Emissions Data System (Hoesly et al., 2018) (Fig. S11). Our inferred wetland emissions in the 2011 La Niña show the highest positive anomaly in the 2010-2013 period, consistent with previous estimates covering this particular period (Pandey et al., 2017). Wetland methane emissions come from anaerobic degradation of organic matter, and hence depend on organic carbon inputs and inundation areas, and logarithmically on temperature (Whalen, 2005). Consistent behaviors between the time and locations of anomalies in the GOSAT $X_{CH_4}$ and changes in wetland extent have been documented with the focus on seasonally flooded wetlands (Parker et al., 2018), but current land models that simulate wetland $CH_4$ emissions have limited skill to capture the dynamics of wetland extent through overbank inundation (Poulter et al., 2017) and they do not quantify stream emissions (Bastviken et al., 2011). An intensification of Amazon flooding extremes is noted according to water levels in the Amazon river, with anomalously high flood levels and long flood durations since 2012 (Barichivich et al., 2018), in line with the inferred wetland $CH_4$ emission increase here.



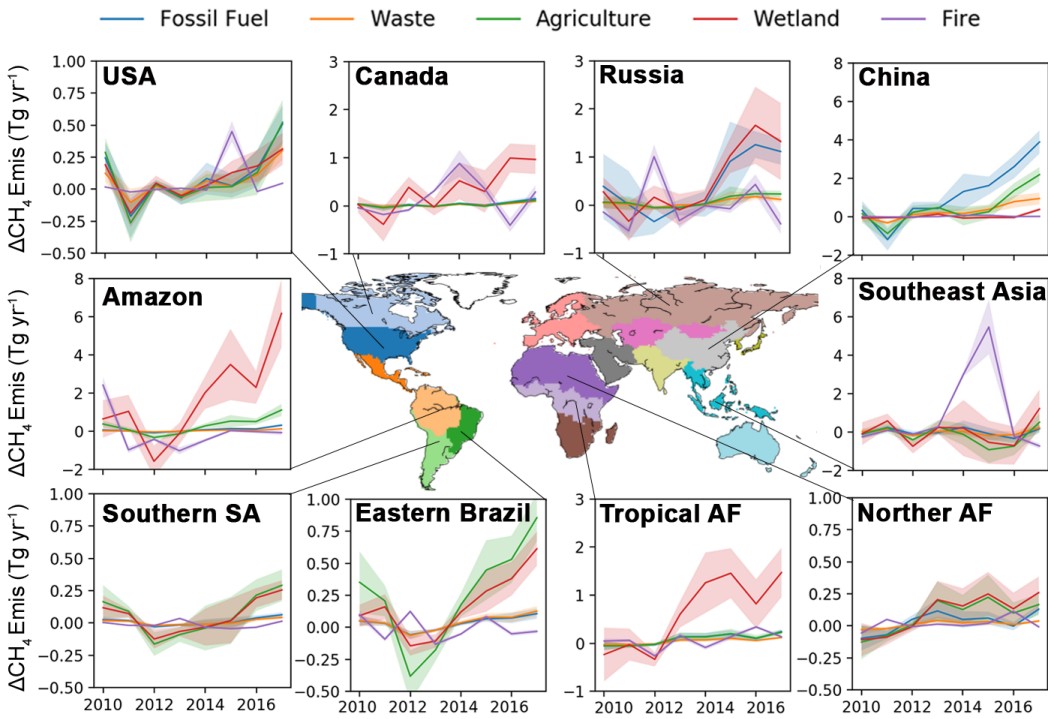

**Figure 8.** Regional emission anomalies relative to the 2010-2013 mean for the sectoral attribution based on prior information. Note the scales on the y-axis are different for each subplot.

For the other tropical regions, significant increases are also attributed to wetland emissions, in particular to Tropical Africa ($1.5\pm0.7$ Tg yr$^{-1}$, Fig. 8), with the largest contribution from the Congo Basin. This attribution is supported by the recent discovery of massive peatlands under the swamp forests (Dargie et al., 2017) and by updated estimates of emissions from African inland waters based on riverine measurements (Borges et al., 2015). Smaller increases are attributed to wetland emissions in the other tropical regions including Eastern Brazil ($0.3\pm0.1$ Tg yr$^{-1}$), Northern Africa ($0.2\pm0.1$ Tg yr$^{-1}$), and Southern South America ($0.1\pm0.1$ Tg yr$^{-1}$). However, other emission sources also play a significant role in these regions, in particular agricultural emissions (Chang et al., 2019). Thus future studies with additional constraints on wetland emissions are needed to better quantify wetland-related changes. Only in Southeast Asia, the major contribution to different CH$_4$ emissions between the two periods is from fire associated with the strong El Niño in 2015 (Yin et al., 2016; Liu et al., 2017). No significant increases are noted for India, consistent with a previous regional study focusing on the 2010-2015 period (Ganesan et al., 2017).

The sectoral breakdown of emissions from China suggests a substantial increase in anthropogenic sources from fossil fuel, agriculture and waste, adding up to an overall trend of $1.0\pm0.2$ Tg yr$^{-2}$ between 2010 and 2017 (Fig. 8). As stated above, this attribution does not account for structural changes in the emission processes, where bottom-up estimates have large uncertainty (Peng et al., 2016). A recent inverse study focusing on Asian emissions from 2010 to 2015 derived nearly the same magnitude





of emission trend for China, and the authors argued that such an increase is likely due to coal mining regardless of recent government regulations, as no significant changes are noted for the other sectors (Miller et al., 2019). A continued increase is confirmed here beyond 2015 till the end of the record in 2017.

      Russia also contributed significant increase in $CH_4$ emissions, by 1.7±0.7 Tg yr$^{-1}$ between 2010-13 and 2014-17 (Fig. 7), possibly from both fossil fuel extraction in Northern Russia and extensive peatland areas (Fig. 8). The surface-driven and
satellite-driven inversions identify slightly different source regions for the rise (Fig. 5). The surface-driven inversions attribute most of the increases to the European part of Russia where anthropogenic emission dominate, whereas the satellite-driven inversions attribute more changes to the West Siberia plain where more wetlands are located (Terentieva et al., 2016). As there are both fossil fuel and wetland sources in the west Siberia plain (Fig. S10), further information is needed to disentangle relative contributions between anthropogenic and natural wetland sources. For the other extratropical regions showing significant $CH_4$
emission increases, the increase in Canada (1.1±0.4 Tg yr$^{-1}$) was mostly attributed to wetlands (Fig. 8), with interannual variations consistent with previous regional inversions (Sheng et al., 2018). Increases in the US (0.7±0.2 Tg yr$^{-1}$) occurred after 2014 with considerable overlapping contributions from different sectors in the prior, preventing a robust sectoral breakdown (Fig. 8).

      Relying on the prior distribution to approximate possible contributions from wetlands in the mid-high latitudes, the in-
crease between 2010-2013 and 2014-2017 amounts to 0.9±0.5, 0.6±0.4, and 0.1±0.06 Tg yr$^{-1}$ for Russia, Canada, and the US. Up to 2012, high-latitude wetland emissions are not identified as significant contributors to increasing atmospheric methane (Saunois et al., 2017). The positive trend in high latitude wetland emissions found here could be the first sign of an impact of the fast warming observed at these latitudes. Adding up all wetland contributions across the globe, changes in wetland emissions dominate the interannual variations in the emission anomaly (Fig. S12a). The general increase in wetland
$CH_4$ fluxes is in line with observed atmospheric $\delta^{13}CH_4$ that shows a general negative trend at all latitudes (Fig. S12b), as biogenic sources like wetlands are more $\delta^{13}CH_4$ depleted than the other ones (Sherwood et al., 2017). The $\delta^{13}CH_4$ changes induced by fossil fuel emission increases (thermogenic) may be balanced by equivalent increases associated with waste and agriculture (biogenic). In particular, two temporal features interrupting the overall decline in $\delta^{13}CH_4$ could be explained by our derived emission anomaly. The negative 2012 wetland emission anomaly, hence a decline in the fraction of $^{13}CH_4$-depleted
sources, coincides with an observed pause in the $\delta^{13}CH_4$ decline in the southern hemisphere. The 2015 positive fire emission anomaly, hence an increase in the fraction of $^{13}CH_4$-rich sources, coincides with the brief reverse of the declining $\delta^{13}CH_4$ in the southern hemisphere. Our attribution is in line with a recent study based on surface $CH_4$ and $\delta^{13}CH_4$ observations, and on a multi-box model to represent zonal emissions (Nisbet et al., 2019), and it moves a step further by identifying key source regions with quantified emission increases. Nevertheless, uncertainties associated with atmospheric inversions need to be better
evaluated through multiple model inter-comparisons. Here, we tested the consistency of different observational constraints and different prior OH distributions. There could be dependencies on the choice of prior emission estimates, and transport model errors could also play a role. In the meantime, future studies using spatial-temporal variations in the observed atmospheric



$\delta^{13}CH_4$ and spatially resolved isotopic source signatures (Ganesan et al., 2018) will provide further constraints on the source attribution.

**5 Conclusions**

Our ensemble of inversions assimilating surface or satellite $CH_4$ observations, as well as chemically-related tracers to partly constrain the OH sink, consistently suggests that the recent acceleration in $CH_4$ growth rate from 2010 to 2017 is most likely induced by increases in surface emissions. The derived global emissions point to an unprecedented new maximum in global total methane emissions. The most substantial increases during the eight study years come from the tropics and East Asia.

Given our prior knowledge on the distribution of different $CH_4$ sources, natural wetland emissions show the largest increase with dominant contributions from the tropics. Such an increase would result in potential positive feedback to climate warming (Zhang et al., 2017). The second-largest increase comes from anthropogenic emissions in China, despite recent government regulations (Miller et al., 2019). The continuation of existing surface $CH_4$ and $\delta^{13}CH_4$ observations and GOSAT/GOSAT-2 $X_{CH_4}$ retrievals, the newly available TROPOspheric Monitoring Instrument (TROPOMI) observations with frequent global

mapping capability (Hu et al., 2018), and the coming of new methane space missions such as the MEthane Remote sensing Lidar missioN (MERLIN) (Bousquet et al., 2018) will bring further insight into regional methane budget changes and their climate sensitivity. At the same time, a process-based understanding of the wetland $CH_4$ emissions and effective anthropogenic emission regulation measures are urgently needed to meet future climate mitigation goals.

*Acknowledgements.* We acknowledge the University of Leicester for the GOSAT $X_{CH_4}$ retrievals, the NCAR MOPITT group for the CO

retrievals, and the Goddard Earth Sciences Data and Information Services Center for the SAO OMI HCHO retrievals. We thank the WD-CGG, NOAA, AGAGE, and TCCON archives to publish the ground-based observations and we are very grateful to all the people involved in maintaining the networks and archiving the data. Specifically, we acknowledge the following networks for making the measurements available: NOAA, CSIRO, ECCC, AGAGE, JMA, UBAG, NIWA, LSCE, MGO, DMC, Empa, FMI, KMA, RSE, SAWS, UMLT, UNIURB, and VNMHA. The Mace Head, Trinidad Head, Ragged Point, Cape Matatula, and Cape Grim AGAGE stations are supported by the National

Aeronautics and Space Administration (NASA) (grants NNX16AC98G to MIT, and NNX16AC97G and NNX16AC96G to SIO). Support also comes from the UK Department for Business, Energy  Industrial Strategy (BEIS) for Mace Head, the National Oceanic and Atmospheric Administration (NOAA) for Barbados, and the Commonwealth Scientific and Industrial Research Organisation (CSIRO) and the Bureau of Meteorology (Australia) for Cape Grim. We also thank F. Marabelle for computing support at LSCE. This work benefited from HPC resources from GENCI-TGCC (Grant 2018-A0050102201). Part of this research was conducted at the NASA sponsored Jet Propulsion Laboratory,

California Institute of Technology, under contract with NASA. This research was also supported by NASA ROSES IDS 80NM0018F0583.



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
