# Peer review of "Accelerating methane growth rate from 2010 to 2017: leading contributions from the tropics and East Asia"

_Atmospheric Chemistry and Physics, 2020_

## Referee Comment (RC1) · Anonymous Referee #1 · 13 Aug 2020

The authors use a range of surface and satellite observations of methane to estimate methane emissions from 2010 to 2017. They also use a combined methane-carbon monoxide-formaldehyde inversion that also uses satellite observations of formaldehyde and carbon monoxide. The study describes a range of calculations that sometimes appear to be cobbled together without any particular logical flow, almost as if two groups have written this without any proper integration. Some of the calculations are also presented in a way that makes it difficult to gain any meaningful insights. The paper would greatly benefit from a robust revision, not least to ensure the authors' key messages are easier to understand. Below I outline my substantive and minor comments.

Substantive comments

Line 54: here (or in methods) it would be useful to outline the caveats associated with the CH4-HCHO-CO method. The method assumes correct knowledge of the underlying chemistry, e.g. the fate of the methyl and higher peroxy radicals.

Line 57: here (or in methods) is an opportunity to tell the readers about any differences in the vertical sensitivity of GOSAT, OMI and MOPITT and how they might impact the combined inversion results. Even if this is addressed in an earlier paper, an acknowledgement would be useful.

Line 63: this reader did not find anywhere in the paper any mention of the ability of this combined system to independently estimate CH4, HCHO and CO.

Section 2.1: by using XCH4 from the proxy retrievals the authors are assuming XCO2. Irrespective of what XCO2 they use, this approach will introduce an error in the posterior emission estimates, which should be acknowledged. The resulting XCH4 data might very well agree within X% of TCCON data but this study is making statements about low and high latitude regions where there is barely any coverage from TCCON.

Line 124: does the optimisation of CH4, CO and HCHO lead to a chemically consistent atmosphere? It would also be useful if the authors reported the methyl chloroform e-folding lifetime as a way of assessing the prior and posterior OH.

Line 139: I was baffled by the diversity of uncertainties attributed to chemical production of HCHO production and OH. Please tell the reader where these values come from. Particularly for the low OH uncertainty, given that later in the study (line 145) the authors explain the large differences between OH fields.

Section 2.2.3: From what this reader understands, the focus of the work is on the 4DVar method. To address the difficulties associated with the ease with which the posterior solution can be characterised using this method, the authors have decided to include additional inversions. This somewhat muddies the water unless the authors

can convincingly show both methods produce consistent emission estimates - not just zonal mean totals. For example, is Figure 6 consistent with the 4DVar system?

Line 209: this is a bold and unsubstantiated statement that appears with no prior warning, e.g. discussion in methods. I am sure the authors could come up with competing reasons for small inter-annual variations.

Line 216: this is a critical point. Later discussions about OH do not appear to address this point.

Line 222: this diversity in results is not addressed very well in the paper and does not bode well for using the alternative set of inversions (section 2.2.3) to help characterise the 4DVar solution. This reader is less concerned about the results using the surface data than the range of results inferred from the satellite data. These satellite inversions are consistent only by virtue of their large uncertainties.

Section 4.1: what I find a bit odd is the authors' use of a four-year period (2010-2013) that includes a La Nina and a subsequent four-year period (2014-2017) that includes a large El Nino. Subtracting these two periods could potentially exaggerate the growth over the eight-year period, particularly over the tropics. Figure S12 shows the temporal changes in global methane emissions (at least I assume it shows the global values). An equivalent figure to accompany Figure 7 would be useful.

Line 288: do Gatti et al and Liu et al use consistent methods to calculate fire emissions? Otherwise, I am unclear how this statement is necessarily valid.

Lines 294-298: this statement does not make sense as written. Are the authors suggesting that variations of XCH4 and wetland extent are consistent but land models that incorporate CH4 emissions are let down by imperfect representations of various hydrological processes? And that is why models do not capture XCH4 variations?

Line 298: the authors' qualitative statement is noted. They noticed a relationship between one study and another. I am certain they can do better than that.

Line 301: tropical African emissions of methane originate mainly from the Congo Basin? That is inconsistent with previous studies. The attribution cannot be "supported" by a statement that large peatlands exist in this region. This reviewer understands from Dargie et al that most of the central part of the basin is permanently flooded in which case why would methane emissions be increasing?

Line 315: another study has estimated Chinese trends in methane are *likely* due to coal mining but is there any evidence in the multi-tracer inversion that this is true? Are the spatial distributions over China consistent with that conclusion? The authors take more time to interpret the Russian signal using spatial distributions. I encourage the authors to do something similar for tropical Africa and China.

Line 342: Hand waving.

Minor comments

Line 41-42: the statements after the first dash makes little sense to this reader.

Line 48: there have been a few studies to investigate the recent acceleration. I urge the authors to use primary references rather than Nisbet et al 2019 reference, which glosses over some of the underlying issues.

Figure caption 1: remove parentheses around Thoning et al.

Line 68: 'We' should be 'we'.

Formaldehyde is referred to as CH2O and HCHO. Please be consistent.

Line 162: posterior or a posteriori. Please be consistent.

Equations 1 to 3: the convention is to use lower case bold for vectors and upper case bold for matrices.

Line 171: reference is a bit mangled.

Line 176: ...generally capture well... This reader fails to understand the meaning of

this statement.

Line 183: anchor points rather than anchoring points?

Line 185: this statement assumes that variations in the column overhead can be related to changes in the underlying surface emissions.

Line 290: typo. Anthropogenic.

Line 294: increase exponentially with temperature?

---

## Referee Comment (RC2) · Anonymous Referee #2 · 4 Sep 2020

This is a very interesting study about a recent increase in the global growth rate of methane and the use of inverse modelling to disentangle the underlying causes. 6 different inversion set ups are used that lead to very consistent results, which is encouraging. The setup of those inversions addresses uncertainties in the treatment of OH, although the results seems to show very little sensitivity to it. Because of this, the added value of CO and CH2O measurements that are used remains unclear. Besides the treatment of OH, some other factors require further attention as will be explained below.

GENERAL COMMENTS

[Figure]

I was surprised to see that the posterior scaling factors for OH remain so close to 1. It is mentioned that low variability is in line with some earlier studies. However, what I am more surprised about is that offsets aren't larger, given the much larger uncertainty in global OH. Looking at figure S6, I see quite a substantial difference in the prior simulation using the two OH fields. It suggests a sizeable difference in the methane lifetime between TRANSCOM and INCA-OH. Surprisingly, this difference does not lead to an OH correction in the inversion, for one are both fields. This suggests, that the updated emissions account for the difference. However, looking at Figure 2, I don't really see a systematic difference between the emissions using the two OH fields either. This must be explained.

The validation presented in the supplement concentrates on CH4, which is fine. However, I was surprised not to see anything about CH2O and CO, and how well inversions 1 and 3 fit those data. This makes it very difficult to judge the performance of these inversion components, and how much we can expect them to influence the estimates for CH4.

It is concluded that the largest contribution to the growth rate increase comes from East Asia and the Tropics. I wonder whether this conclusion may be influenced by the fact that these are also very large fluxes, with large uncertainties. Therefore, you expect the largest adjustments to those fluxes. Suppose the inversion wouldn't know where to put an emission correction. Then the cheapest solution is to distribute it evenly across the globe in terms of fractional deviation from the a prior uncertainty. If that were the case, I suspect that East Asia and the Tropics would stand out also. If East Asia and the Tropics are singled out as main causes explaining the increase, shouldn't that be measured in comparison to this "none-informative" reference rather than absolute emission deviations from the prior?

The supplement provides some evaluation of the inversions against surface and total column data. However, I am missing statistical information on the fits, necessary to judge if the a priori and observational uncertainties are chosen in a realistic and

statistically consistent manner. This information (e.g. chi2) should be provided.

SPECIFIC COMMENTS

Page 1, line 14: I do not think that the '[xx]' notation is correction for representing mixing ratios. In chemistry, the notation is used for concentrations, which is obviously something very different. In my opinion, there should be no confusion between concentration and mixing ratio.

Page 6, line 123: which "meteorological reanalysis"?

Page 6, line 133: Although I understand the rational for using a climatological prior, I nevertheless think it is a problem when investigating the magnitude of trends. Depending on the weight of the prior, the solution will underestimate the trend. As figure S11 confirms, the trend in the climatological prior is biased. Looking at Figure 2b, I get the impression that the trend in the observations is indeed underestimated by the inversion optimized fit. Surprisingly enough the climatological a prior does not affect the a posteriori estimated trend in OH, which I had expected would have accounted for at least part of the missing trend in the posterior solution.

Page 6, line 139: Which information supports the 20% uncertainty in weekly OH per latitude band? I wonder what happens if you integrated the a priori uncertainty in OH globally and per year. The number would probably become very small. Maybe that explains why global mean OH is almost not adjusted in the inversion?

Page 8, line 202: By 'loss rate' you mean 'sink' or 'life time'? I guess 'sink' although 'loss rate' suggest rather 'life time'.

Page 9, line 206: One way to judge how well the inversion is capable to independently estimating the sources and sinks of methane is to look at the posterior correlation between global OH and the global emission. To be able to judge this, it is necessary to provide information on that correlation.

Page 12, line 245: It would be good to refer to Monteil at al (2013), who were the first

to report the difficult to jointly fit surface measurements and GOSAT column retrievals.

Page 13, line 274: Looking at figure S9, I find it hard to be convinced by the argument raised here. For China, the p-value is quite high – so the significance of the positive trend is only low. For the Amazon it looks better. However, I still doubt that it is a good idea to only take the seasonal maximum. It makes the analysis sensitive to extreme events and outliers. Looking at the seasonal coverage a longer common period of data coverage could have been defined. At least some other points should be tried to confirm the robustness of these trends.

Page 13, line 264: The description of regional emission changes is rather silent about the USA. Numerous papers have discussion the increase in fossil fuel related emissions in the past years, potentially explaining a large fraction of the observed global increase in methane. However, I do not see that back in figure 7, which would be worth mentioning.

Page 14, line 275: The difference between OH and emissions that is mentioned here happens by design, since OH is only allowed to be changed in a zonally uniform manner. There is no reason fundamental reason why the sink couldn't change in similar patterns as the source.

Page 17, line 342: Here a connection is made between d13C measurements, and a model analysis that does not account for d13C. Then, how do you know that your results are consistent with d13C? I wonder about the validity of the qualitative arguing in this paragraph. Looking at figure S12, the lags between emission anomalies and d13C responses as well as their amplitudes are difficult to connect between Figures a and b. In reality it is even much more complex due to atmospheric transport variations. Therefore, the way of arguing that it fits together is too easy in my opinion.

Figure 5: What are the small plusses in this figure?

Figure 6: I'm assuming that this figure shows the diagonal of the averaging kernel.

[Figure]

Please mention this somewhere explicitly.

Figure S11: This figure only shows inventory estimated trends. I was surprised not to see the inversion results in the same figure. Since the inventory trends were not used in the a priori, it would be a great way to independently assess the consistency of the inventories and atmospheric data. The fact, that it the posterior fluxes are not included suggests that the comparing might look very good. In either case, some discussion of it is needed. TECHNICAL CORRECTIONS

Page 2, line 21: 'O(1D)' io 'O('D)'

Caption of fig 2: "Deseasoanlized"

Page 15, line 290: "anthropgenic"

Figure 6, caption: 'are shown' io 'is shown'

---

## Author Comment (AC1) · 11 Feb 2021

This is a very interesting study about a recent increase in the global growth rate of methane and the use of inverse modelling to disentangle the underlying causes. 6 different inversion set ups are used that lead to very consistent results, which is encouraging. The setup of those inversions addresses uncertainties in the treatment of OH, although the results seems to show very little sensitivity to it. Because of this, the added value of CO and CH2O measurements that are used remains unclear. Besides the treatment of OH, some other factors require further attention as will be explained below.

We thank the reviewer for the interest in our study and for the constructive and insightful comments that help us to improve our manuscript. Please see our point-to-point responses below.

GENERAL COMMENTS

I was surprised to see that the posterior scaling factors for OH remain so close to 1. It is mentioned that low variability is in line with some earlier studies. However, what I am more surprised about is that offsets aren't larger, given the much larger uncertainty in global OH. Looking at figure S6, I see quite a substantial difference in the prior simulation using the two OH fields. It suggests a sizeable difference in the methane lifetime between TRANSCOM and INCA-OH. Surprisingly, this difference does not lead to an OH correction in the inversion, for one are both fields. This suggests, that the updated emissions account for the difference. However, looking at Figure 2, I don't really see a systematic difference between the emissions using the two OH fields either. This must be explained.

First, we would like to clarify some confusion about Figure 2a. In the original plot, there were two y-axes, with a shift of 20 Tg $CH_4$ emissions per year. Posterior emissions estimated using INCA-OH (shown in circles) are associated with the left y-axis (ranging from 525-575 Tg/yr), while estimates using TransCom-OH are (shown in squares) associated with the right y-axis (ranging from 505-555 Tg/yr). The choice was made to highlight the interannual variations of the posterior $CH_4$ emissions using both OH fields while setting aside the systematic differences. We have updated the plot as shown below to avoid such confusion.

[Figure]

The validation presented in the supplement concentrates on CH4, which is fine. However, I was surprised not to see anything about CH2O and CO, and how well inversions 1 and 3 fit those data. This makes it very difficult to judge the performance of these inversion components, and how much we can expect them to influence the estimates for CH4.

We did not include evaluation data for CO and $CH_2O$ as they were documented in previous papers using the same inverse system (e.g. Yin et al., 2015 and Zheng et al., 2019), and we wanted to keep the focus on $CH_4$ in this study. We agree with the reviewer that such evaluation information would be helpful for the interpretation of the $CH_4$ results, hence we will add evaluation results in the supplementary of the revised manuscript.

It is concluded that the largest contribution to the growth rate increase comes from East Asia and the Tropics. I wonder whether this conclusion may be influenced by the fact that these are also very large fluxes, with large uncertainties. Therefore, you expect the largest adjustments to those fluxes. Suppose the inversion wouldn't know where to put an emission correction. Then the cheapest solution is to distribute it evenly across the globe in terms of fractional deviation from the a prior uncertainty. If that were the case, I suspect that East Asia and the Tropics would stand out also. If East Asia and the Tropics are singled out as main causes explaining the increase, shouldn't that be measured in comparison to this "none-informative" reference rather than absolute emission deviations from the prior?

We agree with the reviewer's insights that the posterior results are regularized by the prior fluxes and associated uncertainties. The ensemble of the 6 inversions included in this study tested uncertainties that stem from observational constraints and OH prior fields, but not from the prior methane emissions. We have added discussion regarding this aspect in the revised manuscript in section 4.3: "We note that the posterior fluxes are regularized by the prior information with climatological fluxes (except for fire) and error statistics that are proportional to the prior fluxes. Nevertheless, our analysis of observational information content using an analytical inverse framework demonstrates that there is information in GOSAT $X_{CH4}$ observations to isolate emissions at continental scales in both mid-latitude and tropics (Fig. 6; Fig. S8), so that the IAV of regionally aggregated fluxes could be well constrained. Furthermore, current bottom-up estimates of both fossil fuel and wetland emissions have large uncertainties (Poulter et al., 2017, Miller et al., 2019, Saunois et al., 2020), introducing uncertain IAV and trend in the prior emissions may degrade the posterior estimates. Therefore, we consider having no prior IAV or trend the best option given that we do not have high confidence in the prior information regarding this aspect. Future follow-up studies that explore uncertainties due to prior information would be very valuable."
We did not compare absolute emission deviations from the prior much in this paper (except for Fig. 4b). As suggested by the reviewer, our paper focused mostly on the temporal changes in the posterior emissions from 2010 to 2017 with each inversion, which avoids direct comparisons of the absolute emission deviations from the prior.

The supplement provides some evaluation of the inversions against surface and total column data. However, I am missing statistical information on the fits, necessary to judge if the a priori

and observational uncertainties are chosen in a realistic and statistically consistent manner. This information (e.g. chi2) should be provided.

We have added statistical information on the fits to the revised manuscript as Supplementary Table 4.

SPECIFIC COMMENTS
Page 1, line 14: I do not think that the '[xx]' notation is correction for representing mixing ratios. In chemistry, the notation is used for concentrations, which is obviously something very different. In my opinion, there should be no confusion between concentration and mixing ratio.

We adopted this notation for its brevity, but we agree with the reviewer's comments. We have revised the manuscript accordingly.

Page 6, line 123: which "meteorological reanalysis"?

ERA-Interim reanalysis. Added in the revised text.

Page 6, line 133: Although I understand the rational for using a climatological prior, I nevertheless think it is a problem when investigating the magnitude of trends. Depending on the weight of the prior, the solution will underestimate the trend. As figure S11 confirms, the trend in the climatological prior is biased. Looking at Figure 2b, I get the impression that the trend in the observations is indeed underestimated by the inversion optimized fit. Surprisingly enough the climatological a prior does not affect the a posteriori estimated trend in OH, which I had expected would have accounted for at least part of the missing trend in the posterior solution.

In the original paper, we stated in section 2.2.2, "This choice is made to avoid prior assumptions about the interannual variations (IAV) or trends in the surface emissions so that IAV in the posterior fluxes are primarily driven by assimilated observations." We have further elaborated on our choice of climatological prior emissions (except for fire) as stated above in our response. As Figure S11 shows, different inventories show diverging trends in many cases, and there are various possible driving factors of the recent methane growth rate as mentioned in the introduction of the manuscript. As such, introducing uncertainty trends or IAV into the prior emission does not seem to be a better option.

For Figure 2b and c, we have added Supplementary Table 5 that summarizes the statistics of the estimated growth rates in the posterior model states against observed ones. Compared to the surface observations, the growth rates of the posterior models are not underestimated. Compared to GOSAT $X_{CH4}$, there are small biases of (-0.33 ppb yr-1) in the surface inversion (S1_TR) that is independent of the GOSAT data.

Table S5. Summary statistics of monthly growth rates comparison between posterior model states and collocated observations as shown in Figure 2b and c. Shaded area indicates that the compared observations are assimilated in the corresponding versions.

| | Compared to Surface Obs (ppb) | | Compared to GOSAT XCH4 (ppb) | |
|---|---|---|---|---|
| | Mean Bias | RMS | Mean bias | RMS |
| $S1_{Surf}\_IN$ | 0.44 | 0.96 | 0.03 | 0.45 |
| $S1_{Surf}\_TR$ | 0.11 | 0.89 | -0.33 | 0.65 |
| $S2_{GOSATonly}\_IN$ | 0.05 | 1.44 | -0.2 | 0.48 |
| $S2_{GOSATonly}\_TR$ | 0.13 | 1.55 | -0.01 | 0.68 |
| $S3_{Multi}\_IN$ | 0.21 | 1.4 | 0.07 | 0.62 |
| $S3_{Multi}\_TR$ | 0.01 | 1.48 | -0.09 | 0.7 |

Both the prior emissions and OH are climatology fields. Given the configuration of the inversion and observed changes in the mixing ratio of the three species, the resultant change in OH is very small and the increases in methane mixing ratio are mostly attributed to changes in surface emissions by the inverse system.

Page 6, line 139: Which information supports the 20% uncertainty in weekly OH per latitude band? I wonder what happens if you integrated the a priori uncertainty in OH globally and per year. The number would probably become very small. Maybe that explains why global mean OH is almost not adjusted in the inversion?

We acknowledge the limitation of such error statistics based on empirical choices. This comment is also related to the previous point that the posterior results are regulated by the prior error statistics. We have added more discussion regarding the underlying caveats in section 2.2.2.

Page 8, line 202: By 'loss rate' you mean 'sink' or 'life time'? I guess 'sink' although 'loss rate' suggest rather 'life time'.

We have corrected it to "the total methane sink".

Page 9, line 206: One way to judge how well the inversion is capable to independently estimating the sources and sinks of methane is to look at the posterior correlation between global OH and the global emission. To be able to judge this, it is necessary to provide information on that correlation.

We agree with the reviewer that it is ideal to have the error covariances of posterior fluxes. However, the computational cost is very expensive with a variational inverse system to estimate the posterior error covariances using a Monte Carlo approach. We have added more discussion regarding this point referencing recent methane inverse studies that use an analytical inversion scheme to optimize anthropogenic methane emissions and their trends on a 4°×5° grid, along with monthly regional wetland emissions and annual hemispheric concentrations of tropospheric OH for individual years (Maasakkers et al., 2019; Zhang et al., 2020). The full error covariances could be computed in such an analytical setting, while the nature of the problem is similar.

Page 12, line 245: It would be good to refer to Monteil at al (2013), who were the first to report the difficult to jointly fit surface measurements and GOSAT column retrievals.

We thank the reviewer for pointing out this important reference. It has been added to the discussion.

Page 13, line 274: Looking at figure S9, I find it hard to be convinced by the argument raised here. For China, the p-value is quite high – so the significance of the positive trend is only low. For the Amazon it looks better. However, I still doubt that it is a good idea to only take the seasonal maximum. It makes the analysis sensitive to extreme events and outliers. Looking at the seasonal coverage a longer common period of data coverage could have been defined. At least some other points should be tried to confirm the robustness of these trends.

We noted in the original manuscript that there are considerable sources of uncertainty for such a gradient analysis. Many factors such as varying sampling in space and time, as well as changes in transport, could result in changes in the latitudinal gradient. Nevertheless, we find this piece of information very interesting. Following the reviewer's comment, we have further highlighted the caveats and the need for more observations.

Page 13, line 264: The description of regional emission changes is rather silent about the USA. Numerous papers have discussion the increase in fossil fuel related emissions in the past years, potentially explaining a large fraction of the observed global increase in methane. However, I do not see that back in figure 7, which would be worth mentioning.

We have added more discussion regarding the lack of trend in the US. "Relatively small increase is found after 2014 with flat emissions before, which is consistent with previous studies finding no trend over US before 2012 (Saunois et al., 2017, Bruhwiler et al., 2017)".

Page 14, line 275: The difference between OH and emissions that is mentioned here happens by design, since OH is only allowed to be changed in a zonally uniform manner. There is no reason fundamental reason why the sink couldn't change in similar patterns as the source.

We would like to clarify that only in S1, where surface $CH_4$ and CO observations are assimilated, the OH fields are optimized in a zonally uniform manner. In Inversions S2 and S3 that assimilate GOSAT $XCH_4$ observations, OH are optimized per each model grid cell. The different choice for surface inversion was made by the limited spatial coverage of surface stations. We have made this point more clear in the revised manuscript.

Page 17, line 342: Here a connection is made between d13C measurements, and a model analysis that does not account for d13C. Then, how do you know that your results are consistent with d13C? I wonder about the validity of the qualitative arguing in this paragraph. Looking at figure S12, the lags between emission anomalies and d13C responses as well as their amplitudes are difficult to connect between Figures a and b. In reality it is even much more complex due to

atmospheric transport variations. Therefore, the way of arguing that it fits together is too easy in my opinion.

We agree with the reviewer on this critique and hence have added $d^{13}C$ simulations with the prior and posterior fluxes using a simple box model.

Figure 5: What are the small plusses in this figure?

They represent trends that are statistically significant at a 95% confidence level. We have added this information in the figure legend.

Figure 6: I'm assuming that this figure shows the diagonal of the averaging kernel. Please mention this somewhere explicitly.

Indeed, the diagonal of the averaging kernel is shown. We have added this information explicitly in the legend.

Figure S11: This figure only shows inventory estimated trends. I was surprised not to see the inversion results in the same figure. Since the inventory trends were not used in the a priori, it would be a great way to independently assess the consistency of the inventories and atmospheric data. The fact, that it the posterior fluxes are not included suggests that the comparing might look very good. In either case, some discussion of it is needed.

We thank the reviewer for this nice suggestion to add the inversion results for comparison. We did not add the inversion results for several reasons: first, it would make the plot very busy so that it becomes difficult to read (adding six versions of inversions); second, this paper does not emphasize the magnitudes of the posterior fluxes compared to the prior (there are systematic zonal differences as illustrated in Fig. 4b), as we would like to focus on the temporal change, adding such a plot would divert the discussion. Figure S11 shows different bottom-up inventory estimates, we have added related discussion in the revised manuscript in section 4.3.

TECHNICAL CORRECTIONS
Page 2, line 21: 'O(1D)' io 'O('D)'
        Corrected.

Caption of fig 2: "Deseasoanlized"
        Corrected.

Page 15, line 290: "anthropgenic"
        Corrected.

Figure 6, caption: 'are shown' io 'is shown'
        Corrected.

---

## Author Response (AR1)

We thank the reviewer's appreciation of our work and the thoughtful comments. We have made corresponding efforts to revise the manuscript. The full review is copied hereafter (in black) and our responses are inserted where appropriate (in blue). Line numbers in the responses are referring to the revised manuscript.

Anonymous Referee #1

The authors use a range of surface and satellite observations of methane to estimate methane emissions from 2010 to 2017. They also use a combined methane-carbon monoxide-formaldehyde inversion that also uses satellite observations of formaldehyde and carbon monoxide. The study describes a range of calculations that sometimes appear to be cobbled together without any particular logical flow, almost as if two groups have written this without any proper integration. Some of the calculations are also presented in a way that makes it difficult to gain any meaningful insights. The paper would greatly benefit from a robust revision, not least to ensure the authors' key messages are easier to understand. Below I outline my substantive and minor comments.

We sincerely thank the reviewer for the constructive comments. We have revised the manuscript accordingly to improve the overall logical flow. Please see responses to individual questions below.

Substantive comments

Line 54: here (or in methods) it would be useful to outline the caveats associated with the CH4-HCHO-CO method. The method assumes correct knowledge of the underlying chemistry, e.g. the fate of the methyl and higher peroxy radicals.

We thank the reviewer for pointing out this point. We have added more discussion regarding model uncertainties in section 2.2.1. "Here, we use a simplified chemistry scheme that assumes methane being oxidized into formaldehyde in a single step. We expect this simplification to have a relatively small impact on the inverse results of methane, given that all pathways of methane oxidation result in formaldehyde as an intermediate product. Besides, HCHO production from non-methane VOC oxidation is simulated upstream with a full-chemistry model, so that the correction on OH from the inversion will not directly feedback to the VOC oxidation. This should not be an issue as we optimize the production of HCHO instead of VOC emissions, but the impact of VOC on OH recycling is not accounted for. Future studies using a full chemistry scheme to optimize methane and OH simultaneously would be helpful to diagnose potential impacts of this simplification on the derived methane lifetime." (line 135-141)

Line 57: here (or in methods) is an opportunity to tell the readers about any differences in the vertical sensitivity of GOSAT, OMI and MOPITT and how they might impact the combined inversion results. Even if this is addressed in an earlier paper, an acknowledgment would be useful.

We have added information regarding the vertical sensitivities of the three satellite retrievals in the method section 2.1.2. "Satellite retrievals of the three species ($CH_4$, HCHO, and CO) we use here are generally sensitive to the entire vertical column with some differences toward the lower troposphere. GOSAT $X_{CH4}$ retrievals using shortwave infrared (SWIR) radiances have approximately uniform sensitivity to methane at all pressure levels (Parker et al., 2015). OMI HCHO retrievals using ultraviolet (UV) radiance are sensitive to the entire column with some decline in the lowest atmospheric layers (Gonzalez Abad et al., 2015). For MOPITT, we use the multispectral total column CO retrieval products that combine near-infrared (NIR) and thermal infrared (TIR) radiances and hence have an enhanced sensitivity to the lower troposphere (Deeter et al., 2014). Such subtle differences in the vertical sensitivities of the three retrievals as well as their different vertical profiles and lifetimes may influence the ways the observations of the three species inform about OH, which is another source of uncertainty in addition to the model and observation errors." (line 113-121)

Line 63: this reader did not find anywhere in the paper any mention of the ability of this combined system to independently estimate CH4, HCHO and CO.

This information is documented in section 2.2.1. "This inversion system has been documented and evaluated by a series of studies focusing on tracers including $CH_4$ (Pison et al., 2009; Locatelli et al., 2015; Cressot et al., 2014), HCHO (Fortems-Cheiney et al., 2012), CO (Yin et al., 2015; Zheng et al., 2019), and $CO_2$ (Chevallier et al., 2005, 2010)." (line 130-132)

Section 2.1: by using XCH4 from the proxy retrievals the authors are assuming XCO2. Irrespective of what XCO2 they use, this approach will introduce an error in the posterior emission estimates, which should be acknowledged. The resulting XCH4 data might very well agree within X% of TCCON data but this study is making statements about low and high latitude regions where there is barely any coverage from TCCON.

We agree with the reviewer that there are measurement errors irrespective of the retrieval methods. We have expanded the part about observation uncertainty in section 2.1.2 to provide more details regarding retrieval errors. "Here, we use GOSAT $X_{CH4}$ proxy retrievals (OCPR) version 7.2 from the University of Leicester, which has been well documented and evaluated against various observations. The retrieval has a single-observation precision of ~14 ppb (~0.7%) and a

regional bias of ~4 ppb compared to TCCON stations (Parker et al., 2015, 2020). This product is also consistent with other GOSAT methane retrievals (Buchwitz et al., 2017). However, we note that there is limited spatial coverage of TCCON stations to fully evaluate GOSAT observations in the high-latitudes and the tropics. " (line 100-104)

Line 124: does the optimisation of CH4, CO and HCHO lead to a chemically consistent atmosphere? It would also be useful if the authors reported the methyl chloroform e-folding lifetime as a way of assessing the prior and posterior OH.

The optimization does not change much of the prior OH field (INCA or TransCom) (Fig. 3), both of which have been documented by previous studies such as Patra et al., 2011, Naik et al., 2013, and Zhao et al., 2020 that are referenced in the paper.

Line 139: I was baffled by the diversity of uncertainties attributed to chemical production of HCHO production and OH. Please tell the reader where these values come from. Particularly for the low OH uncertainty, given that later in the study (line 145) the authors explain the large differences between OH fields.

The uncertainty assigned to OH (20%) was low compared to that of the HCHO production (200%). Previous studies found relatively small interannual variations in OH (Montzka et al., 2011; Nicely et al., 2018), and a relatively small prior uncertainty was used in many inverse studies (e.g. 10% on the hemispheric mean and 5% on individual years in Zhang et al., 2021).  As for the large uncertainty of HCHO production, it accounts for error propagations from VOC emission estimates (based on MEGAN2.0) to the HCHO production simulated by LMDz-INCA. Future studies exploring different characteristics of the prior error statistics will be very valuable.

Section 2.2.3: From what this reader understands, the focus of the work is on the 4DVar method. To address the difficulties associated with the ease with which the posterior solution can be characterised using this method, the authors have decided to include additional inversions. This somewhat muddies the water unless the authors can convincingly show both methods produce consistent emission estimates - not just zonal mean totals. For example, is Figure 6 consistent with the 4DVar system?

There are trade-offs between a variational inverse system and an analytical one. A variational system can handle large state vectors and hence our system can account for multiple species simultaneously (i.e. CH4-HCHO-CO), and at the same time, effectively reduce aggregation errors in both space and time (i.e. optimize gridded fluxes or scaling factors on a weekly basis). However, it cannot estimate the averaging kernels of the atmospheric inversion that describe the sensitivity of the posterior solution to the true fluxes, as well as the error covariances of the posterior. An analytical system can estimate this useful diagnostic information, but it is limited by

computational capacity so that some temporal and regional aggregations of the fluxes are needed to construct the state vector such that the response functions are dependent on the spatial-temporal pattern of the prior fluxes for each state vector.

Our major point here is to estimate the information content of available observations from surface stations or the GOSAT satellites to inform regional methane fluxes. As the available observations, the prior fluxes, and the transport models are the same, these estimates using the simple analytical inverse system (as shown in Figure 6) can provide relevant information to help us interpret the inversion results derived from the variational system that optimizes gridded fluxes of the three species simultaneous.

Line 209: this is a bold and unsubstantiated statement that appears with no prior warning, e.g. discussion in methods. I am sure the authors could come up with competing reasons for small inter-annual variations.

We have revised the text to make a proper transition from results to the discussion. "The resultant small interannual variations in the posterior OH field is in line with a modeling study that showed a high OH recycling probability and hence a weak sensitivity to emission perturbations (Lelieveld et al., 2016)." (line 231-233)

Line 216: this is a critical point. Later discussions about OH do not appear to address this point.

We have revised the text to layout the caveats, "Still, we cannot rule out the possibility that numerically it might be easier for the optimization system to adjust surface emissions of the three species to fit the observations rather than modifying OH to adjust the sink terms in the absence of a mechanistic chemical feedback in the chemical transport model. The feedback effects are mostly tested using box models at the current stage (Prather et al., 1994, Nguyen et al., 2020), future studies accounting for these effects in a 3-D inversion would be helpful to diagnose its impacts on estimated changes in methane lifetimes." (line 238-242).

Line 222: this diversity in results is not addressed very well in the paper and does not bode well for using the alternative set of inversions (section 2.2.3) to help characterise the 4DVar solution. This reader is less concerned about the results using the surface data than the range of results inferred from the satellite data. These satellite inversions are consistent only by virtue of their large uncertainties.

For the inversions that assimilate GOSAT $X_{CH4}$ observations, there are systematic differences in the resultant zonal emission magnitudes due to differences in (1) prior OH field and (2) GOSAT only or multi-species constraints as shown in Figure 4b. Nevertheless, they all agree on the interannual variations as shown in Figure 4a, which is what this paper primarily focuses on.  The

differences between S2 (GOSAT only) and S3 (Multi-species) given the same OH are relatively small in most of the zones except for the 0-30N band, which is mainly due to the scaling of OH in S3 as informed by HCHO and CO observations.

Section 4.1: what I find a bit odd is the authors' use of a four-year period (2010-2013) that includes a La Nina and a subsequent four-year period (2014-2017) that includes a large El Nino. Subtracting these two periods could potentially exaggerate the growth over the eight-year period, particularly over the tropics. Figure S12 shows the temporal changes in global methane emissions (at least I assume it shows the global values). An equivalent figure to accompany Figure 7 would be useful.

As suggested by the reviewer, annual changes in regional emissions are shown in Figure 8, while differences between the 2010-2013 and 2014-2017 period are shown in Figure 7 using the same regional mask. We have made the connections more apparent in the revised text. We agree with the reviewer that such La Nina and El Nino contrast may influence our interpretation about methane growth over the eight years. We have added some discussion regarding this point in section 4.3. "Looking into regional emission changes, the differences in the posterior $CH_4$ emissions between the last and the first four years of our study period (2014-2017 vs. 2010-2013) are shown in Fig. 7, while regional masks of the 18 sub-regions and regional annual emission anomalies are shown in Fig. 8." "This increase does not necessarily imply linear trends in emissions as there are considerable interannual variations in the derived emissions (Fig. 8), in particular, the first period includes a La Niña year 2011 during which high tropical wetland emissions have been reported (Pandey et al., 2017) and the latter period includes a strong El Nino year 2015 during which large fire emissions from indonesia have been reported (Yin et al., 2016; Worden et al., 2017)." (line 288-295)

Line 288: do Gatti et al and Liu et al use consistent methods to calculate fire emissions? Otherwise, I am unclear how this statement is necessarily valid.

The two studies used different observations and approaches (Gatti et al. primarily used aircraft campaigns and Liu et al. used satellite retrievals of CO and $CO_2$). They both showed the same temporal changes in fire emissions that are in line with our findings here, which we consider as independent observational support of our results.

Lines 294-298: this statement does not make sense as written. Are the authors suggesting that variations of XCH4 and wetland extent are consistent but land models that incorporate CH4 emissions are let down by imperfect representations of various hydrological processes? And that is why models do not capture XCH4 variations?

We have removed this comment following the reviewer's suggestion.

Line 298: the authors' qualitative statement is noted. They noticed a relationship between one study and another. I am certain they can do better than that.

Detailed analysis of regional drivers on methane emission changes from wetland would be an interesting follow up study. However, it may exceed the scope of the current paper. We have revised the text as "An intensification of Amazon flooding extremes has been documented based on water levels in the Amazon river, with anomalously high flood levels and long flood durations since 2012 (Barichivich et al., 2018}, which could result in higher wetland $CH_4$ emissions." (line 320-322)

Line 301: tropical African emissions of methane originate mainly from the Congo Basin? That is inconsistent with previous studies. The attribution cannot be "supported" by a statement that large peatlands exist in this region. This reviewer understands from Dargie et al that most of the central part of the basin is permanently flooded in which case why would methane emissions be increasing?

We concur with the reviewer and have revised the text accordingly. "Our result of increasing tropical Africa wetland emissions is consistent with a recent regional inversion using GOSAT data at a high spatial resolution of 0.5°×0.625°, which find a positive trend of 1.5–2.1 Tg yr$^{-2}$ in the region over 2010 to 2016, mainly attributed to wetlands in the Sudd in South Sudan (Lunt et al., 2019)." (line 324-326)

Line 315: another study has estimated Chinese trends in methane are *likely* due to coal mining but is there any evidence in the multi-tracer inversion that this is true? Are the spatial distributions over China consistent with that conclusion? The authors take more time to interpret the Russian signal using spatial distributions. I encourage the authors to do something similar for tropical Africa and China.

We agree with the reviewer's comments and have revised the discussion accordingly to include most recent publications. "The sectoral breakdown of emissions from China suggests a substantial increase in anthropogenic sources from fossil fuel, agriculture and waste, adding up to an overall trend of 1.0±0.2 Tg yr$^{-2}$ between 2010 and 2017 (Fig. 8). As stated above, this attribution relies on the relative contribution of different sectors from the prior information and does not account for structural changes in time. A recent inverse study focusing on Asian emissions from 2010 to 2015 derived nearly the same magnitude of emission trend for China (Miller et al., 2019), a continued increase is confirmed here beyond 2015 till the end of the record in 2017. In contrast, a global inversion that separated the mean anthropogenic emissions and trends in the state vector found a smaller trend in anthropogenic emissions over China (0.39±0.27 Tg yr$^{-2}$) for the period 2010-2018, and a trend of 0.72±0.39 Tg yr$^{-2}$ focusing on the

period 2010-2016 (Zhang et al., 2021). The numbers are comparable given the differences in the inverse setups and the chemical transport models being used." (line 333-341)

Minor comments

Line 41-42: the statements after the first dash makes little sense to this reader.

We have deleted it following the reviewer's suggestion.

Line 48: there have been a few studies to investigate the recent acceleration. I urge the authors to use primary references rather than Nisbet et al 2019 reference, which glosses over some of the underlying issues.

We thank the reviewer for this suggestion and have added more references including McNorton et al., 2018, Turner et al., 2019, Zhang et al., 2021.

Figure caption 1: remove parentheses around Thoning et al.

Corrected.

Line 68: 'We' should be 'we'.

Corrected.

Formaldehyde is referred to as CH2O and HCHO. Please be consistent.

Thanks for pointing this out. All changed to HCHO.

Line 162: posterior or a posteriori. Please be consistent.

All changed to posterior for consistency.

Equations 1 to 3: the convention is to use lower case bold for vectors and upper case bold for matrices.

Corrected.

Line 171: reference is a bit mangled.

Corrected.

Line 176: ...generally capture well. . . This reader fails to understand the meaning of this statement.

Revised as "In general, the global average methane growth rate is well captured by the posterior model states"

Line 183: anchor points rather than anchoring points?

Corrected.

Line 185: this statement assumes that variations in the column overhead can be related to changes in the underlying surface emissions.

The statement here compares the measurement precision and spatial coverage (including vertical sampling) between the two types of observations without discussion or assumptions about underlying surface emissions.

Line 290: typo. Anthropogenic.
Corrected.

Line 294: increase exponentially with temperature?

Yes. Changed accordingly.

Anonymous Referee #2

This is a very interesting study about a recent increase in the global growth rate of methane and the use of inverse modelling to disentangle the underlying causes. 6 different inversion set ups are used that lead to very consistent results, which is encouraging. The setup of those inversions addresses uncertainties in the treatment of OH, although the results seems to show very little sensitivity to it. Because of this, the added value of CO and CH2O measurements that are used remains unclear. Besides the treatment of OH, some other factors require further attention as will be explained below.

We thank the reviewer for the interest in our study and for the constructive comments. Please see our point-to-point responses below.

GENERAL COMMENTS

I was surprised to see that the posterior scaling factors for OH remain so close to 1. It is mentioned that low variability is in line with some earlier studies. However, what I am more surprised about is that offsets aren't larger, given the much larger uncertainty in global OH. Looking at figure S6, I see quite a substantial difference in the prior simulation using the two OH fields. It suggests a sizeable difference in the methane lifetime between TRANSCOM and INCA-OH. Surprisingly, this difference does not lead to an OH correction in the inversion, for one are both fields. This suggests, that the updated emissions account for the difference. However, looking at Figure 2, I don't really see a systematic difference between the emissions using the two OH fields either. This must be explained.

First, we would like to clarify some confusion about Figure 2a. In the original plot, there were two y-axises, with a shift of 20 Tg $CH_4$ emissions per year. Posterior emissions estimated with INCA-OH (shown in circles) are associated with the left y-axis (ranging from 525-575 Tg/yr), while posterior emissions estimated with TransCom-OH are (shown in squares) associated with the right y-axis (ranging from 505-555 Tg/yr). The choice was made to highlight the interannual variations of the posterior $CH_4$ emissions using both OH fields, instead of the systematic differences. We have updated the plot as shown below for clarity.

[Figure]

The validation presented in the supplement concentrates on CH4, which is fine. However, I was surprised not to see anything about CH2O and CO, and how well inversions 1 and 3 fit those data. This makes it very difficult to judge the performance of these inversion components, and how much we can expect them to influence the estimates for CH4.

We did not include evaluation data for CO and $CH_2O$ as they were documented in previous papers in Yin et al., 2015, Yin et al., 2017 and Zheng et al., 2019.

It is concluded that the largest contribution to the growth rate increase comes from East Asia and the Tropics. I wonder whether this conclusion may be influenced by the fact that these are also very large fluxes, with large uncertainties. Therefore, you expect the largest adjustments to those fluxes. Suppose the inversion wouldn't know where to put an emission correction. Then the cheapest solution is to distribute it evenly across the globe in terms of fractional deviation from the a prior uncertainty. If that were the case, I suspect that East Asia and the Tropics would stand out also. If East Asia and the Tropics are singled out as main causes explaining the increase, shouldn't that be measured in comparison to this "none-informative" reference rather than absolute emission deviations from the prior?

We agree with the insights of the reviewer that the posterior results are regularized by the a priori fluxes and associated uncertainties. However, the assumption that "Suppose the inversion wouldn't know where to put an emission correction" assumes that the geographical distribution of $X_{CH4}$ is uninformative, which does not hold here given the information content analysis we show in Figure 6. The averaging kernels of the GOSAT constraints are higher than 0.85 in many regions, suggesting that regional emission changes are well captured given available

observations. For the remaining cross-error terms, we agree that future studies exploring uncertainties due to different prior information would be very valuable.

The supplement provides some evaluation of the inversions against surface and total column data. However, I am missing statistical information on the fits, necessary to judge if the a priori and observational uncertainties are chosen in a realistic and statistically consistent manner. This information (e.g. chi2) should be provided.

We have added this information to the revised manuscript. "The reduced chi-squared (J divided by the number of observations) is about 0.5, which is much lower than 1 because of observation error inflation to compensate for the fact that we do not account for observation error correlations following findings of Chevallier et al., (2007)." (line 149-151)

SPECIFIC COMMENTS
Page 1, line 14: I do not think that the '[xx]' notation is correction for representing mixing ratios. In chemistry, the notation is used for concentrations, which is obviously something very different. In my opinion, there should be no confusion between concentration and mixing ratio.

We adopted this notation for its brevity, but we agree that it results in unnecessary confusion. We have revised the manuscript accordingly.

Page 6, line 123: which "meteorological reanalysis"?

ERA-Interim reanalysis. Added in the revised text.

Page 6, line 133: Although I understand the rational for using a climatological prior, I nevertheless think it is a problem when investigating the magnitude of trends. Depending on the weight of the prior, the solution will underestimate the trend. As figure S11 confirms, the trend in the climatological prior is biased. Looking at Figure 2b, I get the impression that the trend in the observations is indeed underestimated by the inversion optimized fit. Surprisingly enough the climatological a prior does not affect the a posteriori estimated trend in OH, which I had expected would have accounted for at least part of the missing trend in the posterior solution.

We stated in the manuscript, "This choice is made to avoid prior assumptions about the interannual variations (IAV) or trends in the surface emissions so that IAV in the posterior fluxes are primarily driven by assimilated observations." As the reviewer pointed out earlier, the posterior fluxes are influenced by the prior fluxes, which would impact the derived trends and

IAV of the posterior fluxes. Therefore, a climatology prior was preferred (except for fire emissions).

Page 6, line 139: Which information supports the 20% uncertainty in weekly OH per latitude band? I wonder what happens if you integrated the a priori uncertainty in OH globally and per year. The number would probably become very small. Maybe that explains why global mean OH is almost not adjusted in the inversion?

We acknowledge the limitation of such error statistics based on empirical choices. We have added more discussion regarding the underlying caveats in section 3.3.

Page 8, line 202: By 'loss rate' you mean 'sink' or 'life time'? I guess 'sink' although 'loss rate' suggest rather 'life time'.

We have corrected it to "the total methane sink".

Page 9, line 206: One way to judge how well the inversion is capable to independently estimating the sources and sinks of methane is to look at the posterior correlation between global OH and the global emission. To be able to judge this, it is necessary to provide information on that correlation.

We agree with the reviewer that it is ideal to have the error covariances of posterior fluxes. However, the computational cost is very expensive with a variational inverse system to estimate the posterior error covariances using a Monte Carlo approach. We have added more discussion regarding this point citing methane inverse studies optimizing emissions and global OH simultaneously using GOSAT observations with an analytical inversion scheme (Maasakkers et al., 2019; Zhang et al., 2021). "Recent GOSAT inverse studies explored optimizing gridded annual anthropogenic methane emissions and associated trends, regional monthly wetland emissions, and global (or hemispheric) annual OH concentration with an analytical inversion scheme, where it is possible to compute the full posterior error covariance matrix (Maasakkers et al., 2019; Zhang et al., 2021). The results suggest a strong negative error correlation between global anthropogenic emissions and methane lifetime (r=-0.8), moderate correlations between wetland emissions and methane lifetime (r=-0.4), and between OH trend and wetland or anthropogenic emission trends (r=-0.6) (Zhang et al., 2021). Hence, assimilating GOSAT data alone, the inversion has limited information to separate the sources and the sinks.  With our multi-species variational inverse system, it is computational too costly to estimate the posterior error covariances using a Monte Carlo approach. Given the strong error correlations between the source and sink terms identified by Zhang et al., (2021), we cannot rule out the possibility that numerically it might be

easier for the optimization system to adjust surface emissions of the three species to fit the observations rather than modifying OH to adjust the sink terms in the absence of a mechanistic chemical feedback in the chemical transport model. The feedback effects are mostly tested using box models at the current stage (Prather et al., 1994; Nguyen et al., 2020}, future studies accounting for these effects in a 3-D inversion would be helpful to diagnose its impacts on estimated changes in methane lifetimes. " (line 239-251)

Page 12, line 245: It would be good to refer to Monteil at al (2013), who were the first to report the difficult to jointly fit surface measurements and GOSAT column retrievals.

We thank the reviewer for pointing out this reference. It has been added to the discussion.

Page 13, line 274: Looking at figure S9, I find it hard to be convinced by the argument raised here. For China, the p-value is quite high – so the significance of the positive trend is only low. For the Amazon it looks better. However, I still doubt that it is a good idea to only take the seasonal maximum. It makes the analysis sensitive to extreme events and outliers. Looking at the seasonal coverage a longer common period of data coverage could have been defined. At least some other points should be tried to confirm the robustness of these trends.

We noted in the original manuscript that there are considerable sources of uncertainty for such a gradient analysis. Many factors such as varying sampling in space and time, as well as changes in transport, could result in changes in the latitudinal gradient. Nevertheless, we find this piece of information interesting. Given the reviewer's comment, we have removed it from the supplementary information.

Page 13, line 264: The description of regional emission changes is rather silent about the USA. Numerous papers have discussion the increase in fossil fuel related emissions in the past years, potentially explaining a large fraction of the observed global increase in methane. However, I do not see that back in figure 7, which would be worth mentioning.

We have added more discussion regarding the lack of trend in the US. "Relatively small increase is found after 2014 with flat emissions before, which is consistent with previous studies finding no trend over US before 2012 (Saunois et al., 2017, Bruhwiler et al., 2017)".

Page 14, line 275: The difference between OH and emissions that is mentioned here happens by design, since OH is only allowed to be changed in a zonally uniform manner. There is no reason fundamental reason why the sink couldn't change in similar patterns as the source.

We would like to clarify that only in S1, where surface CH$_4$ and CO observations are assimilated, the OH fields are optimized in a zonally uniform manner. In Inversions S2 and S3 that assimilate GOSAT XCH$_4$ observations, OH are optimized per each model grid cell. The different choice for surface inversion was made by the limited spatial coverage of surface stations. We have made this point more clear in the revised manuscript.

Page 17, line 342: Here a connection is made between d13C measurements, and a model analysis that does not account for d13C. Then, how do you know that your results are consistent with d13C? I wonder about the validity of the qualitative arguing in this paragraph. Looking at figure S12, the lags between emission anomalies and d13C responses as well as their amplitudes are difficult to connect between Figures a and b. In reality it is even much more complex due to atmospheric transport variations. Therefore, the way of arguing that it fits together is too easy in my opinion.

We agree with the reviewer on this critique and hence have removed this qualitative discussion.

Figure 5: What are the small plusses in this figure?

They represent trends that are statistically significant at a 95% confidence level. We have added this information in the figure legend.

Figure 6: I'm assuming that this figure shows the diagonal of the averaging kernel. Please mention this somewhere explicitly.

Indeed, the diagonal of the averaging kernel is shown. We have added this information explicitly in the legend.

Figure S11: This figure only shows inventory estimated trends. I was surprised not to see the inversion results in the same figure. Since the inventory trends were not used in the a priori, it would be a great way to independently assess the consistency of the inventories and atmospheric data. The fact, that it the posterior fluxes are not included suggests that the comparing might look very good. In either case, some discussion of it is needed.

We thank the reviewer for this nice suggestion, however, adding inversion results from the six versions would make the plot too busy to read.

TECHNICAL CORRECTIONS
Page 2, line 21: 'O(1D)' io 'O('D)'

Corrected.

Caption of fig 2: "Deseasoanlized"
Corrected.

Page 15, line 290: "anthropgenic"
Corrected.

Figure 6, caption: 'are shown' io 'is shown'
Corrected.

---

## Author Response (AR2)

A few points remain that have not sufficiently been addressed in my opinion:

We thank the reviewer for the comments and have addressed the remaining questions below.

>>The choice was made to highlight the interannual variations of the posterior CH4 emissions using both OH fields, instead of the systematic differences. We have updated the plot as shown below for clarity.

But if the inversion is not sensitive to the mean OH level, I wonder what can be expected for its IAV. In my opinion it is still important to discuss the insensitivity of the inversion to the mean difference between the a priori OH fields.

We concur with the reviewer on this point. We have added in the manuscript that "The inversion adjusted surface emission levels given the two different prior OH fields (Fig. 2), indicating that there is not enough information to constrain the magnitudes of the sources and sinks of the three species separately with their atmospheric observations." (Line 257-259). Followed by more discussions in Section 3.3.

>>We stated in the manuscript, "This choice is made to avoid prior assumptions about the interannual variations (IAV) or trends in the surface emissions so that IAV in the posterior fluxes are primarily driven by assimilated observations." As the reviewer pointed out earlier, the posterior fluxes are influenced by the prior fluxes, which would impact the derived trends and IAV of the posterior fluxes. Therefore, a climatology prior was preferred (except for fire emissions).

The choice has been well motivated, but not the possible consequence of using a biased prior for the results that are obtained.

We have added Supplementary Table 4 to summarize the statistical comparison of growth rates in the posterior model states against the observed ones. The posterior model states captured the observed growth rate reasonably well, therefore the worry that "Depending on the weight of the prior, the solution will underestimate the trend" is not justified.

Table S4. Summary statistics of monthly growth rates comparison between posterior model states and collocated observations as shown in Figure 2b and c. Shaded area indicates that the compared observations are assimilated in the corresponding versions.

|                        | Compared to Surface Obs (ppb yr -1 ) |      | Compared to GOSAT XCH4 (ppb yr -1 ) |      |
|------------------------|-------------------------------------------------|------|------------------------------------------------|------|
|                        | Mean Bias                                       | RMS  | Mean bias                                      | RMS  |
| SI Surf_ IN | 0.44                                            | 0.96 | 0.03                                           | 0.45 |
| S1 Surf_ TR | 0.11                                            | 0.89 | -0.33                                          | 0.65 |

| S2 GOSATonly _IN | 0.05 | 1.44 | -0.2  | 0.48 |
|-----------------------------|------|------|-------|------|
| $S2_{GOSATonly}_TR$         | 0.13 | 1.55 | -0.01 | 0.68 |
| S3 Multi _IN     | 0.21 | 1.4  | 0.07  | 0.62 |
| S3 Multi _TR     | 0.01 | 1.48 | -0.09 | 0.7  |

>>We thank the reviewer for this nice suggestion, however, adding inversion results from the six versions would make the plot too busy to read.

With a little more creativity a solution to this problem would have been found. In this case at least some discussion about this comparison should have been added in my opinion. The author's response confirms my suspicion that the comparison raises questions.

We thank the reviewer for the suggestion. We have added our inversion results into this supplementary figure S11 as shown below. In each subplot, the solid lines represent the six inversion results of this study, while the dashed lines represent bottom-up inventories that are different from the ones used in our prior (shown in S3 and S9 instead). The comparison does not raise specific questions.

We note that there are no observational constraints for the sectoral attribution of the total fluxes. Only the total methane emissions are optimized at each model grid, and the sectoral attribution relies solely on the prior knowledge. As climatology priori fluxes are used for most sectors except for fire emissions, no temporal changes in the source structures of the other sectors are accounted for. Hence, such attribution only serves as a step to understand the dominant sources of each region. The different magnitudes of posterior fluxes among the six inversions stem from systematic differences between surface and satellite observational constraints, as well as different prior OH fields (discussed in Section 4.1).